# Squared families are useful conjugate priors

**Russell Tsuchida**
Data Science and AI Group
Monash University
russell.tsuchida@monash.edu

**Jiawei Liu**
School of Computing
Australian National University
jiawei.liu3@anu.edu.au

**Cheng Soon Ong**
Data61, CSIRO
and Australian National University
chengsoon.ong@anu.edu.au

**Dino Sejdinovic**
RAIR, AIML
The University of Adelaide
dino.sejdinovic@adelaide.edu.au

## Abstract

Squared families of probability distributions have been studied and applied in numerous machine learning contexts. Typically, they appear as likelihoods, where their advantageous computational, geometric and statistical properties are exploited for fast estimation algorithms, representational properties and statistical guarantees. Here, we investigate the use of squared families as prior beliefs in Bayesian inference. We find that they can form helpful conjugate families, often allowing for closed-form and tractable Bayesian inference and marginal likelihoods. We apply such conjugate families to Bayesian regression in feature space using end-to-end learnable neural network features. Such a setting allows for a rich multi-modal alternative to Gaussian processes with neural network features, often called deep kernel learning. We demonstrate our method on few shot learning, outperforming existing neural methods based on Gaussian processes and normalising flows.[1]

## 1 Introduction

**Squared families are useful likelihoods** Families $\{q(\cdot \mid \boldsymbol{f})\}_{\boldsymbol{f} \in \mathcal{H}}$ of probability densities of the form $q(\cdot \mid \boldsymbol{f}) \propto \|\boldsymbol{f}(\cdot)\|_2^2$, where $\boldsymbol{f} \in \mathcal{H}$ and $\mathcal{H}$ is some suitably nice space of vector-valued functions, have recently emerged as promising likelihoods in several different communities in machine learning. For example, functions of the form $\boldsymbol{f}(\cdot) = \boldsymbol{\Theta}\boldsymbol{\psi}(\cdot)$ for some $\boldsymbol{\Theta} \in \mathbb{R}^{m \times n}$ and $\boldsymbol{\psi}(\cdot) = \left(k(\cdot, \boldsymbol{a}_1), \ldots, k(\cdot, \boldsymbol{a}_N)\right)^{\top}$, where $(\boldsymbol{a}_i)_{i=1}^N$ are data points and $k$ is the kernel of a reproducing kernel Hilbert space (RKHS), are obtained as minimisers of regularised empirical risk problems [Marteau-Ferey et al., 2020]. When appropriately normalised and applied to divergence minimisation, they can be used to model probability densities [Rudi and Ciliberto, 2021]. When $m = 1$, such density models can also be applied to modelling intensities (of Poisson point processes) [Flaxman et al., 2017], and Bayesian variants which use Gaussian processes (GPs) instead of RKHS models are also readily available [Walder and Bishop, 2017]. Separately to kernel methods, probabilistic circuits [Choi et al., 2020] can be chosen for $\boldsymbol{f}$, which can be squared [Sladek, 2023, Loconte et al., 2023a,b, Loconte and Vergari, 2025], or squared and summed [Loconte et al., 2024], to obtain tractable and expressive models for probability density functions. Finally, neural networks can be used for $\boldsymbol{f}$, and when they are two-layered and fully-connected, they often admin tractable and closed-form normalising constants, marginal distributions and conditional distributions [Tsuchida et al., 2023].

---

[1]Code available at https://github.com/Carlisle-Liu/SNEFY-Process.git.

39th Conference on Neural Information Processing Systems (NeurIPS 2025).

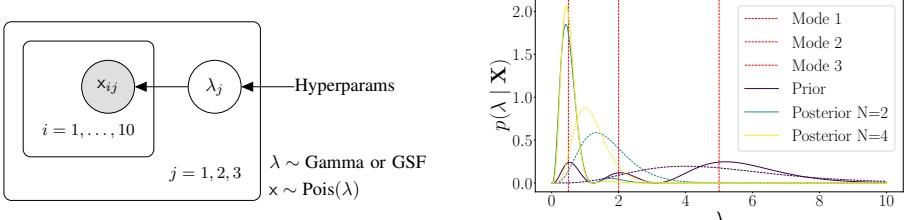

Figure 1: (Left) Empirical Bayesian estimation of the rate parameter $\lambda$ of a Poisson likelihood in a hierarchical model. Here the mode index $j = 1, 2, 3$ is observed, but not the mode rate $\lambda_j$. A classical empirical Bayesian approach might be to place a shared conjugate prior over $\lambda_j$ (in this case, the Gamma density, see Appendix A), and maximise the marginal likelihood with respect to the prior hyperparameters. Classical conjugate family priors are only capable of expressing a unimodal prior belief, unless they are mixture models. (Right) At inference time, we are presented $N$ datapoints from a single mode — Mode 1. GSF priors (solid curves) are multimodal, and are able to adapt their trimodal prior to the observations quickly, whereas the unimodal Gamma prior (dashed curves) have to slowly adapt their density to pick out the correct mode. GSFs significantly generalise mixture models of simpler base conjugate priors.

**Are squared families useful priors?**   In this work, we examine the use of such families as priors, and more specifically, families of conjugate priors. To the best of our knowledge, we are the first to consider such families as priors. Note that this question is distinct from a Bayesian treatment of squared family likelihoods [Walder and Bishop, 2017, for example], where a squared family is used for a likelihood and some prior belief over $\boldsymbol{f}$ (for example, $\boldsymbol{f} \sim \mathcal{GP}$) is employed. Rather, this question is about using some other (not necessarily squared family likelihood) and a squared family prior. We first construct a hierarchy of families, called generalised squared families (GSF). We then show when a prior belongs to the GSF, the marginal likelihoods, posterior parameter and posterior predictive distributions all admit closed-form expressions. Furthermore, the posterior distributions remain within the GSF. This allows for rich and multimodal expression of prior beliefs within a conjugate family (see Figure 1, for example). We apply our conjugate families to the problem of Bayesian regression in feature space, which offers a model called GSF processes (GSFP) with rich multi-modal uncertainty and capabilities well-suited for few-task learning.

## 2   Background

**Conjugate priors**   Let $\Pi$ be a family of probability measures each supported on a subset of $\Omega$. We call $\Pi$ a *conjugate class* for a likelihood $p(\boldsymbol{u} \mid \boldsymbol{\omega})$ if for all $\pi \in \Pi$, the posterior probability measure $\pi'$ defined by $\pi'(d\boldsymbol{\omega} \mid \boldsymbol{u}) = p(\boldsymbol{u} \mid \boldsymbol{\omega})\pi(d\boldsymbol{\omega}) / \int_\Omega p(\boldsymbol{u} \mid \boldsymbol{\omega}')\pi(d\boldsymbol{\omega}')$ is also an element of $\Pi$. An immediate but unhelpful conjugate class is the class $\mathbb{Q}$ of all probability measures, since for any likelihood a prior in $\mathbb{Q}$ results in a posterior in $\mathbb{Q}$ [Robert, 2007, page 98]. Often when authors speak of conjugate families, they implicitly and loosely also require that such conjugate families have closed-form or tractable normalising constants or parameters. In order to facilitate such closed-form updates, conjugate families are therefore required to be small and analytically tractable. Exponential families of priors matched to exponential family likelihoods are arguably the most well-known examples of conjugate priors (see Robert [2007, §3.3.3] and Appendix A). We explicitly differentiate between conjugate families and conjugate families with closed-form normalising constants. We will consider two distinct types of conjugate families: one for classes of squared families, and one for their corresponding base measures. Both conjugacies are with respect to the same likelihood function.

**Gaussian processes**   Gaussian processes (GPs) are flexible nonparametric models for probabilistic inference over functions. Typically one specifies that some function of interest a priori follows a GP, takes some observations from the likelihood given the value of the GP, and then computes the posterior conditioned on the observations under the likelihood. If the likelihood is Gaussian, then the posterior process is also a GP because the Gaussian distribution is a conjugate prior for the Gaussian likelihood. There are multiple ways to mathematically construct GPs, two of which are the function space view and the weight space view [Rasmussen and Williams, 2006, § 2.2 and § 2.1].

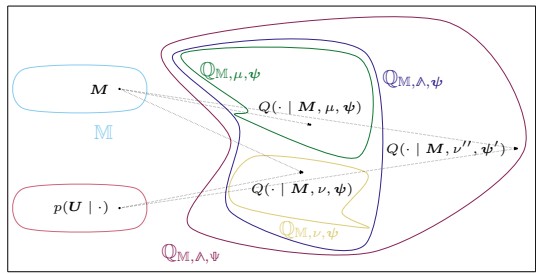

Figure 2: Given base measure $\mu$ and feature $\boldsymbol{\psi}$, a parameter $\boldsymbol{M} \in \mathbb{M}$ indexes a prior $Q(\cdot \mid \boldsymbol{M}, \mu, \boldsymbol{\psi})$ from a generalised squared family (GSF) $\mathbb{Q}_{\mathbb{M}, \mu, \boldsymbol{\psi}}$ (see Definition 1). Given some observed data $\boldsymbol{U}$ and likelihood $p(\boldsymbol{U} \mid \cdot)$, the same parameter $\boldsymbol{M} \in \mathbb{M}$ indexes the posterior $Q(\cdot \mid \boldsymbol{M}, \nu, \boldsymbol{\psi})$, where $\nu = p(\boldsymbol{U} \mid \cdot)\mu(\cdot)$ (see Proposition 3). The posterior and prior both belong to the conjugate class $\mathbb{Q}_{\boldsymbol{M}, \mathbb{A}, \boldsymbol{\psi}}$, a GSF (see Definition 1). In certain cases of linear regression in feature space, where the prior is placed over coefficients and the likelihood is Gaussian with expectation described by the regression function, the same parameter $\boldsymbol{M} \in \mathbb{M}$ indexes the posterior predictive distribution $Q(\cdot \mid \boldsymbol{M}, \nu'', \boldsymbol{\psi}')$, for some appropriate $\nu''$ and $\boldsymbol{\psi}'$ (see Corollary 6). The posterior predictive, posterior parameter, and prior distributions all belong to the conjugate class $\mathbb{Q}_{\boldsymbol{M}, \mathbb{A}, \Psi}$, also a GSF. Closed-form expressions for all densities in the GSF $\mathbb{Q}_{\mathbb{M}, \mathbb{A}, \Psi}$ as well as the marginal likelihood (see Proposition 2) allow for joint learning of $\boldsymbol{M}$, $\mu$ and $\boldsymbol{\psi}$ of the prior distribution via type II maximum likelihood, or empirical Bayes (see § 4), and for closed-form posterior updates.

**Function-space view**   A (vector valued) Gaussian process $\{\mathbf{f}(\boldsymbol{x})\}_{\boldsymbol{x} \in \mathbb{X}}$, or simply $\mathbf{f}$, is a collection of (possibly infinitely many) random $d_2$-dimensional vectors such that every finite subset $\{\mathbf{f}(\boldsymbol{x}_i)\}_{i=1}^{N}$ follows an $Nd_2$-variate Gaussian distribution. A GP is characterised by its vector-valued mean function $m : \mathbb{X} \to \mathbb{R}^{d_2}$ and positive semidefinite PSD matrix-valued covariance (or kernel) function $\boldsymbol{K} : \mathbb{X} \times \mathbb{X} \to \mathbb{S}_+^{d_2}$, where $\mathbb{S}_+^{d_2}$ is the set of $d_2 \times d_2$ PSD matrices. We write $\mathbf{f} \sim \mathcal{GP}(\boldsymbol{m}, \boldsymbol{K})$ to mean that $\mathbf{f}$ is a GP with mean and covariance functions $\boldsymbol{m}$ and $\boldsymbol{K}$ respectively, meaning that for any finite set of indices $\boldsymbol{X} = \{\mathbf{x}_i\}_{i=1}^{N}$, $\text{vec}\big(\mathbf{f}(\boldsymbol{X})\big) \in \mathbb{R}^{Nd_2}$ has mean matrix $\text{vec}\big(\boldsymbol{m}(\boldsymbol{X})\big) \in \mathbb{R}^{Nd_2}$ and block partitioned covariance matrix with $ij$th block $\boldsymbol{K}(\boldsymbol{x}_i, \boldsymbol{x}_j) \in \mathbb{S}_+^{d_2}$.

**Weight-space view**   Finite-feature GPs can be constructed by specifying a feature mapping $\boldsymbol{\gamma}$, defining a function $\mathbf{f}(\boldsymbol{x}) = \boldsymbol{\gamma}(\boldsymbol{x})\boldsymbol{\Omega}^{\top}$ and choosing a Gaussian distribution over the parameters $\boldsymbol{\Omega}$. Let $\boldsymbol{\Omega}^{\top} \in \mathbb{R}^{d_1 \times d_2}$ follow a matrix Gaussian distribution such that $\text{vec}(\boldsymbol{\Omega}^{\top})$ has mean zero and covariance $\boldsymbol{B} \otimes \boldsymbol{C}$ with $\boldsymbol{B} \in \mathbb{S}_+^{d_2}$ and $\boldsymbol{C} \in \mathbb{S}_+^{d_1}$, with the notation $\otimes$ denoting the Kronecker product. Let $\boldsymbol{\gamma} : \mathbb{X} \to \mathbb{R}^{d_1}$ be a finite feature mapping and define $\boldsymbol{\Gamma} = \boldsymbol{\gamma}(\boldsymbol{X})$. Since Gaussian random vectors are closed under addition, one may verify that $\text{vec}\big(\mathbf{f}(\boldsymbol{X})\big) = \text{vec}\big(\boldsymbol{\Gamma}\boldsymbol{\Omega}^{\top}\big)$ follows a $Nd_2$-variate Gaussian distribution. The covariance matrix is given by

$$\mathbb{E}\big[\text{vec}(\boldsymbol{\Gamma}\boldsymbol{\Omega}^{\top})\,\text{vec}(\boldsymbol{\Gamma}\boldsymbol{\Omega}^{\top})^{\top}\big] = \boldsymbol{B} \otimes (\boldsymbol{\Gamma}\boldsymbol{C}\boldsymbol{\Gamma}^{\top}).$$

We note that this is a special-case way of handling vector-valued GPs, corresponding with the so-called coregionalisation model [Alvarez et al., 2012]. Our later discussions likely extend to more involved vector-valued setups, with more involved notations.

**Conditionals and marginals of GPs**   One attractive property of GPs is that marginal, conditional and evaluations of linear operators of GPs are also GPs. This in particular implies that if one uses a GP function prior or Gaussian parameter prior and one makes observations from a Gaussian likelihood, the posterior parameter distribution (if applicable) and the posterior predictive distribution are also Gaussian and available in closed-form. Furthermore, the marginal likelihood is available in closed-form and is Gaussian. Access to the marginal likelihood allows for empirical Bayes, or type II maximum likelihood for estimation of hyperparameters of the kernel and mean functions. When combined with deep learning, this approach is known as deep kernel learning [Wilson et al., 2016]. This allows for deep neural network features to be used end-to-end in probabilistic regression models. Unfortunately, while expressive deep learning features are used, which leads to improved point estimates, the uncertainty itself is limited to a unimodal Gaussian distribution.

**Generalised squared families**  Let $\boldsymbol{\psi} : \Omega \to \mathbb{R}^n$ be some feature mapping, and let $\mu$ be a measure supported on $\Omega$. We construct densities $q(\boldsymbol{\omega} \mid \boldsymbol{\Theta}, \mu, \boldsymbol{\psi})$ (with respect to measure $\mu$) which are proportional to the squared Euclidean norm of $\boldsymbol{\Theta}\boldsymbol{\psi}(\boldsymbol{\omega})$. That is,

$$q(\boldsymbol{\omega} \mid \boldsymbol{\Theta}, \mu, \boldsymbol{\psi}) = \frac{\left\|\boldsymbol{\Theta}\boldsymbol{\psi}(\boldsymbol{\omega})\right\|_2^2}{z(\boldsymbol{\Theta})}, \qquad z(\boldsymbol{\Theta}) = \int_\Omega \left\|\boldsymbol{\Theta}\boldsymbol{\psi}(\boldsymbol{\omega})\right\|_2^2 d\mu(\boldsymbol{\omega}).$$

The normalising constant $z(\boldsymbol{\Theta})$ satisfies a convenient *parameter-integral* factorisation

$$z(\boldsymbol{\Theta}) = \mathrm{Tr}\left(\boldsymbol{\Theta}^\top \boldsymbol{\Theta}\Big(\overbrace{\int_\Omega \boldsymbol{\psi}(\boldsymbol{\omega})\boldsymbol{\psi}(\boldsymbol{\omega})^\top \mu(d\boldsymbol{\omega})}^{\boldsymbol{K}_{\mu,\boldsymbol{\psi}}\triangleq}\Big)\right),$$

which follows from the cyclic property of the trace and linearity of the integral. The *squared family kernel* $\boldsymbol{K}_{\mu,\boldsymbol{\psi}}$ is a PSD matrix, as is $\boldsymbol{\Theta}^\top \boldsymbol{\Theta}$, and the normalising constant may be viewed as an inner product of PSD matrices. The squared family kernel is available in closed-form for many combinations of $\Omega$, $\mu$ and $\boldsymbol{\psi}$, with examples from the neural network Gaussian process [Neal, 1995] and neural tangent kernel literature [Han et al., 2022, table 1], random feature literature [Rahimi and Recht, 2007], and classical exponential families [Nielsen and Garcia, 2024] (see Appendix B). The parameter factor $\boldsymbol{\Theta}^\top \boldsymbol{\Theta}$ involves no integration, and the integral factor $\boldsymbol{K}_{\mu,\boldsymbol{\psi}}$ does not depend on any parameters. This factorisation extends beyond the normalising constant to other integrals (such as those involved in the marginal likelihood, posterior parameter, and posterior predictive distributions) and is very helpful, allowing factorisation into $\boldsymbol{\Theta}^\top \boldsymbol{\Theta}$ and a parameter-independent integral.

The numerator $\left\|\boldsymbol{\Theta}\boldsymbol{\psi}(\boldsymbol{\omega})\right\|_2^2 = \mathrm{Tr}\left(\boldsymbol{\Theta}^\top \boldsymbol{\Theta}\boldsymbol{\psi}(\boldsymbol{\omega})\boldsymbol{\psi}(\boldsymbol{\omega})^\top\right)$ may also be understood as an inner product of a PSD parameter matrix and a rank 1 feature matrix. Hence we may also parameterise $q(\boldsymbol{\omega} \mid \boldsymbol{\Theta}, \mu, \boldsymbol{\psi})$ and its corresponding probability measure $Q(d\boldsymbol{\omega} \mid \boldsymbol{M}, \mu, \boldsymbol{\psi})$ in terms of $\boldsymbol{M} = \boldsymbol{\Theta}^\top \boldsymbol{\Theta}$,

$$Q(d\boldsymbol{\omega} \mid \boldsymbol{M}, \mu, \boldsymbol{\psi}) = \frac{\mathrm{Tr}\left(\boldsymbol{M}\boldsymbol{\psi}(\boldsymbol{\omega})\boldsymbol{\psi}(\boldsymbol{\omega})^\top\right)}{z(\boldsymbol{M})}\mu(d\boldsymbol{\omega}), \qquad z(\boldsymbol{M}) = \mathrm{Tr}\left(\boldsymbol{M}\boldsymbol{K}_{\mu,\boldsymbol{\psi}}\right). \qquad (1)$$

Distributions of the form (1) generalise mixture models, which are obtained when $\boldsymbol{M}$ is diagonal.

## 3   Conjugacy of Generalised Squared Families

In order to better describe convenient properties of Bayesian updates, we introduce various families of probability densities. We later describe precisely how in some sense these families are closed under Bayesian updating, and how this closure is computationally attractive.

**Definition 1.** *Let $\mathbb{M}$ be a set of $n \times n$ PSD matrices, $\mathbb{A}$ be a set of nonnegative measures, and $\Psi$ be a set of feature mappings of the form $\boldsymbol{\psi} : \Omega \to \mathbb{R}^n$. A generalised squared family (GSF) is a set of probability measures of the form (1),*

$$\mathbb{Q}_{\mathbb{M},\mathbb{A},\Psi} = \{Q(d\boldsymbol{\omega} \mid \boldsymbol{M}, \mu, \boldsymbol{\psi})\}_{\boldsymbol{M} \in \mathbb{M}, \mu \in \mathbb{A}, \boldsymbol{\psi} \in \Psi},$$

*indexed by $\boldsymbol{M} \in \mathbb{M}, \mu \in \mathbb{A}, \boldsymbol{\psi} \in \Psi$. We allow any of the indexing sets $\mathbb{M}, \mathbb{A}, \Psi$ to be a singleton, and with an abuse of notation, write the singleton's element in place of the set. For example,*

$$\mathbb{Q}_{\boldsymbol{M},\mathbb{A},\boldsymbol{\psi}} = \{Q(d\boldsymbol{\omega} \mid \boldsymbol{M}, \mu, \boldsymbol{\psi})\}_{\mu \in \mathbb{A}} \quad and \quad \mathbb{Q}_{\boldsymbol{M},\mathbb{A},\Psi} = \{Q(d\boldsymbol{\omega} \mid \boldsymbol{M}, \mu, \boldsymbol{\psi})\}_{\mu \in \mathbb{A}, \boldsymbol{\psi} \in \Psi}.$$

We note that by Von Neumann's trace inequality, $\mathbb{M}$ may be taken to be the set of all non-zero PSD matrices as long as the squared family kernel $\boldsymbol{K}_{\mu,\boldsymbol{\psi}}$ is PD, since $\mathrm{Tr}\left(\boldsymbol{M}\boldsymbol{K}_{\mu,\boldsymbol{\psi}}\right) = z(\boldsymbol{M}) > 0$. The inclusions $\mathbb{Q}_{\mathbb{M},\mu,\boldsymbol{\psi}} \subseteq \mathbb{Q}_{\mathbb{M},\mathbb{A},\boldsymbol{\psi}} \subseteq \mathbb{Q}_{\mathbb{M},\mathbb{A},\Psi}$, together with a summary of some of our later results, are visualised in Figure 2. Previous work [Tsuchida et al., 2025] called families of the form $\mathbb{Q}_{\mathbb{M},\mu,\boldsymbol{\psi}}$, where $\mathbb{M}$ is the set of rank 1 PSD matrices, squared families. Throughout this paper, we use $Q$ for probability measures belonging to GSFs, and $P$ and $p$ for arbitrary probability measures and densities (which may also belong to GSFs, where stated).

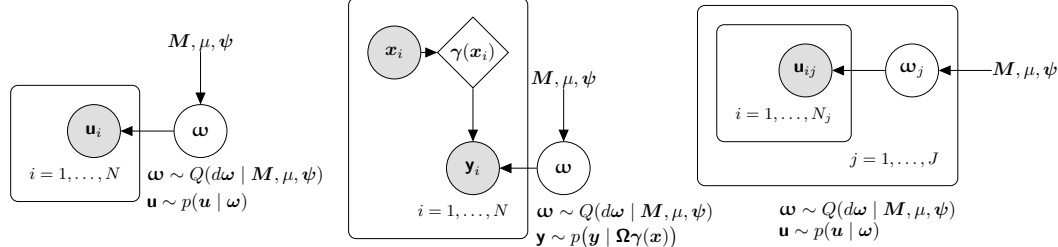

Figure 3: (Left) In § 3.1, we consider an abstract model where random variables following GSFs $\boldsymbol{\omega}$ are latent under an observation model $\mathbf{u} \sim p(\boldsymbol{u} \mid \boldsymbol{\omega})$. (Middle) In § 3.2 specialise on a setting where we expand the observed node $\mathbf{u}$ so that observations take the form of $(\boldsymbol{x}, \boldsymbol{y})$ pairs, and assume an observation model of the form $\mathbf{y} \sim p(\boldsymbol{y} \mid \boldsymbol{\Omega}\boldsymbol{\gamma}(\boldsymbol{x}))$. Here $\boldsymbol{\omega} = \mathrm{vec}(\boldsymbol{\Omega}^\top)$ is a parameter and $\boldsymbol{\gamma}$ is some deterministic feature mapping (for example, from a deep neural network). Finally in § 3.3, we focus on the setting where the likelihood $p(\boldsymbol{y} \mid \boldsymbol{\Omega}\boldsymbol{\gamma}(\boldsymbol{x}))$ and base measure $\mu$ are both Gaussian, allowing us to generalise finite-feature Gaussian process regression. (Right) Our results extend to the case of a hierachical model with a shared prior over $J$ datasets. We apply this to few-shot learning in § 4.

## 3.1 Marginal likelihood and posterior distributions

We first consider an abstract setting where there are two variables of interest — a variable which we observe and a latent variable which we do not observe but would like to infer. The latent variable $\boldsymbol{\omega}$ a priori is distributed $\boldsymbol{\omega} \sim Q(d\boldsymbol{\omega} \mid \boldsymbol{M}, \mu, \boldsymbol{\psi})$ according to an element of a GSF, and is mapped to the observed variable $\mathbf{U} = \{\mathbf{u}_i\}_{i=1}^N$ implicitly through some arbitrary likelihood function $p(\boldsymbol{u} \mid \boldsymbol{\omega})$. This model is depicted in the left part of Figure 3.

The main trick in performing posterior updates in GSFs is that the base measure and the likelihood combine to give a new base measure. We derive the marginal likelihood in the main text to expose the simplicity of this idea. Here we make use of the parameter-integral factorisation. The proofs of later results follow a similar manipulation (see Appendices C, D).

**Proposition 2.** *Consider the left model of Figure 3. Define a measure $\nu(d\boldsymbol{\omega}) = p(\boldsymbol{U} \mid \boldsymbol{\omega})\mu(d\boldsymbol{\omega})$ as the product of the likelihood and base measure. The marginal likelihood is the ratio of normalising constants, $p(\boldsymbol{U}) = \mathrm{Tr}\left(\boldsymbol{M}\boldsymbol{K}_{\nu,\boldsymbol{\psi}}\right) / \mathrm{Tr}\left(\boldsymbol{M}\boldsymbol{K}_{\mu,\boldsymbol{\psi}}\right)$ (where $\boldsymbol{K}_{\nu,\boldsymbol{\psi}}$ depends on $\boldsymbol{U}$ through $\nu$).*

*Proof.* The marginal likelihood $p(\boldsymbol{U})$ is given by

$$\int_\Omega p(\boldsymbol{U} \mid \boldsymbol{\omega})Q(d\boldsymbol{\omega} \mid \boldsymbol{M}, \mu, \boldsymbol{\psi}) = \frac{\mathrm{Tr}\left(\boldsymbol{M} \int_\Omega \boldsymbol{\psi}(\boldsymbol{\omega})\boldsymbol{\psi}(\boldsymbol{\omega})^\top \overbrace{p(\boldsymbol{U} \mid \boldsymbol{\omega})\mu(d\boldsymbol{\omega})}^{\nu(d\boldsymbol{\omega})}\right)}{\mathrm{Tr}\left(\boldsymbol{M}\boldsymbol{K}_{\mu,\boldsymbol{\psi}}\right)} = \frac{\mathrm{Tr}\left(\boldsymbol{M}\boldsymbol{K}_{\nu,\boldsymbol{\psi}}\right)}{\mathrm{Tr}\left(\boldsymbol{M}\boldsymbol{K}_{\mu,\boldsymbol{\psi}}\right)}.$$

$\square$

The marginal likelihood does not belong to a GSF. [2] The posterior distribution does belong to a GSF, even for finite $N$. We may form a GSF which includes both the prior and the posterior, with an updated base measure and an unchanged parameter.

**Proposition 3.** *Consider the model in Figure 3 (Left). Define a measure $\nu(d\boldsymbol{\omega}) = p(\boldsymbol{U} \mid \boldsymbol{\omega})\mu(d\boldsymbol{\omega})$. The posterior $P(d\boldsymbol{\omega} \mid \boldsymbol{U}) = Q(d\boldsymbol{\omega} \mid \boldsymbol{M}, \nu, \boldsymbol{\psi})$ belongs to a GSF with base measure $\nu$, feature mapping $\boldsymbol{\psi}$ and parameter $\boldsymbol{M}$.*

---

[2] Informally, while the marginal likelihood itself does not belong to a GSF, it asymptotically belongs to a GSF, since the measure $\nu$ is proportional to a posterior under prior $\mu$. Therefore under mild conditions, by the Bernstein-von Mises theorem, $\nu$ concentrates to a Dirac delta distribution centered at the MLE $\hat{\boldsymbol{\omega}}_N$. Hence $p(\boldsymbol{U})/\int \nu(d\boldsymbol{\omega}) \approx \mathrm{Tr}\left(\boldsymbol{M}\boldsymbol{\psi}(\hat{\boldsymbol{\omega}}_N)\boldsymbol{\psi}(\hat{\boldsymbol{\omega}}_N)^\top\right) / \mathrm{Tr}\left(\boldsymbol{M}\boldsymbol{K}_{\mu,\boldsymbol{\psi}}\right)$. This would allow previous results on maximum likelihood estimation with GSFs [Tsuchida et al., 2025] to extend to maximum marginal likelihood estimation. We leave rigorous investigation of this observation for future work.

Proposition 3 implies that the GSF $\mathbb{Q}_{M,\mathbb{A},\psi}$ is a conjugate class for likelihood $p(\boldsymbol{u} \mid \boldsymbol{\omega})$, where $\mathbb{A}$ is the set of all measures that can be expressed in the form $\left(\prod_{i=1}^{h} p(\boldsymbol{u}_i \mid \boldsymbol{\omega})\right)\mu(d\boldsymbol{\omega})$ for some integer $h \geq 0$ and $\boldsymbol{u}_i$ belonging to the support of the likelihood. We note that the posterior predictive is $p(\boldsymbol{U}_* \mid \boldsymbol{U})$ is also available, since it is essentially of the same form as the marginal likelihood, replacing the GSF prior with a GSF posterior and the training likelihood with the test likelihood.

**Proposition 4.** *Consider the model in Figure 3 (Left). Define a measure* $\nu(d\boldsymbol{\omega}) = p(\boldsymbol{U} \mid \boldsymbol{\omega})\mu(d\boldsymbol{\omega})$ *and* $\nu'(d\boldsymbol{\omega}) = p(\boldsymbol{U}_* \mid \boldsymbol{\omega})\nu(d\boldsymbol{\omega})$, *where, abusing notation, the train likelihood* $p(\boldsymbol{U} \mid \boldsymbol{\omega})$ *and test likelihood* $p(\boldsymbol{U}_* \mid \boldsymbol{\omega})$ *are not necessarily the same. The posterior predictive is the ratio of normalising constants,* $p(\boldsymbol{U}_* \mid \boldsymbol{U}) = \operatorname{Tr}\left(\boldsymbol{M}\boldsymbol{K}_{\nu',\psi}\right)/\operatorname{Tr}\left(\boldsymbol{M}\boldsymbol{K}_{\nu,\psi}\right)$.

The following observation illustrates that it is easy to obtain closed-form expressions for the marginal likelihood, posterior parameter and posterior predictive distributions, whenever the base measure is itself conjugate to the likelihood (for exponential family examples, see Appendix A).

**Remark 5.** *If* $\mathbb{A}$ *is a conjugate class for* $p(\boldsymbol{u} \mid \boldsymbol{\omega})$, *then* $\nu \in \mathbb{A}$ *(up to a normalising constant), so if the squared family kernel* $\boldsymbol{K}_{\mu,\psi}$ *is available in closed-form for all* $\mu \in \mathbb{A}$, *then so is* $\boldsymbol{K}_{\nu,\psi}$.

Note that we here refer to two different conjugate updates: One as if the prior were base measure $\mu$ belonging to $\mathbb{A}$ and another for the true prior belonging to a GSF $\mathbb{Q}_{M,\mathbb{A},\psi}$. As described in § 2 and Appendix B, closed-form expressions for the squared family kernel are available for many of combinations of $\Omega$, $\mu$ and $\psi$ [Neal, 1995, Williams, 1996, Rahimi and Recht, 2007, Cho and Saul, 2009, Nielsen and Garcia, 2024, Han et al., 2022, Tsuchida et al., 2023, for example].

## 3.2 Regression models

We now consider a more specific instance of the previously described model, as shown in the middle of Figure 3. We consider Bayesian learning of vector-valued functions of the form

$$\mathbf{f}(\boldsymbol{x}) = \boldsymbol{\Omega}\boldsymbol{\gamma}(\boldsymbol{x}), \qquad \boldsymbol{\omega} \sim Q(d\boldsymbol{\omega} \mid \boldsymbol{M}, \mu, \psi), \boldsymbol{\omega} = \operatorname{vec}(\boldsymbol{\Omega}^\top) \in \mathbb{R}^d, \quad \text{where} \quad d = d_1 d_2, \quad (2)$$

where $\boldsymbol{\Omega}^\top \in \mathbb{R}^{d_1 \times d_2}$ are readout parameters and $\boldsymbol{\gamma} : \mathbb{X} \to \mathbb{R}^{d_1}$ is a feature mapping (from e.g. a deep neural network). We assume our observations are independent samples $(\boldsymbol{X}, \boldsymbol{Y}) = \{(\boldsymbol{x}_i, \boldsymbol{y}_i)\}_{i=1}^N$ from $p\big(\boldsymbol{y} \mid \boldsymbol{\Omega}\boldsymbol{\gamma}(\boldsymbol{x})\big)$ and some distribution over $\mathbf{x}$. Let $\boldsymbol{\Gamma} = \boldsymbol{\gamma}(\boldsymbol{X})$. The marginal likelihood,

$$p(\boldsymbol{Y} \mid \boldsymbol{\Gamma}) = \int_\Omega p(\boldsymbol{Y} \mid \boldsymbol{\Gamma}\boldsymbol{\Omega}^\top)Q(d\boldsymbol{\omega} \mid \boldsymbol{M}, \mu, \psi),$$

serves two purposes. First, as a normalisation constant for the posterior, and second, as an objective function for point estimation of $\boldsymbol{M}$, $\mu$, and $\psi$. The posterior parameter distribution is given by

$$P(d\boldsymbol{\Omega} \mid \boldsymbol{Y}, \boldsymbol{\Gamma}) = \frac{p(\boldsymbol{Y} \mid \boldsymbol{\Gamma}\boldsymbol{\Omega}^\top)Q(d\boldsymbol{\omega} \mid \boldsymbol{M}, \mu, \psi)}{p(\boldsymbol{Y} \mid \boldsymbol{\Gamma})},$$

which in turn induces a posterior over noisy evaluations $\boldsymbol{Y}_*$ of $\mathbf{f}$ at new test points (features) $\boldsymbol{X}_*$ ($\boldsymbol{\Gamma}_*$). This posterior is called the (noisy) posterior predictive distribution,

$$p\big(\boldsymbol{Y}_* \mid \boldsymbol{Y}, \boldsymbol{\Gamma}, \boldsymbol{\Gamma}_*\big) = \int_\Omega p(\boldsymbol{Y}_* \mid \boldsymbol{\Gamma}_*\boldsymbol{\Omega}^\top)P(d\boldsymbol{\Omega} \mid \boldsymbol{Y}, \boldsymbol{\Gamma}),$$

and is used to form predictions on test data $\boldsymbol{X}_*$. Note that the test likelihood $p(\boldsymbol{Y}_* \mid \boldsymbol{\Gamma}_*\boldsymbol{\Omega}^\top)$ may be different to the train likelihood $p(\boldsymbol{Y} \mid \boldsymbol{\Gamma}\boldsymbol{\Omega}^\top)$, e.g. with heteroskedastic noise. The noiseless posterior predictive $p(\mathbf{f}(\boldsymbol{X}_*) \mid \boldsymbol{Y}, \boldsymbol{\Gamma}, \boldsymbol{\Gamma}_*)$ is obtained whenever the test likelihood has zero variance.

**Marginal likelihood, posterior parameter, and posterior predictive distributions** The results of Propositions 2, 3 and 4 apply directly to the marginal likelihood, posterior parameter and posterior predictive distributions. That is, defining $\nu(d\boldsymbol{\omega}) = p(\boldsymbol{Y} \mid \boldsymbol{\Gamma}\boldsymbol{\Omega}^\top)\mu(d\boldsymbol{\omega})$ and $\nu'(d\boldsymbol{\omega}) = p(\boldsymbol{Y}_* \mid \boldsymbol{\Gamma}_*\boldsymbol{\Omega}^\top)\nu(d\boldsymbol{\omega})$, the marginal likelihood is $p(\boldsymbol{Y} \mid \boldsymbol{\Gamma}, \boldsymbol{M}) = \operatorname{Tr}\left(\boldsymbol{M}\boldsymbol{K}_{\nu,\psi}\right)/\operatorname{Tr}\left(\boldsymbol{M}\boldsymbol{K}_{\mu,\psi}\right)$, the posterior parameter distribution $P(d\boldsymbol{\Omega} \mid \boldsymbol{Y}, \boldsymbol{\Gamma}) = Q(d\boldsymbol{\omega} \mid \boldsymbol{M}, \nu, \psi)$ belongs to a GSF and the posterior predictive distribution is $p(\boldsymbol{Y}_* \mid \boldsymbol{\Gamma}_*, \boldsymbol{Y}, \boldsymbol{\Gamma}) = \operatorname{Tr}\left(\boldsymbol{M}\boldsymbol{K}_{\nu',\psi}\right)/\operatorname{Tr}\left(\boldsymbol{M}\boldsymbol{K}_{\nu,\psi}\right)$. Note also that Remark 5 still applies; if $\mu, \nu$ and $\nu'$ remain inside the same conjugate class and $\boldsymbol{K}_{\cdot,\psi}$ is available within that family, then closed-form normalising constants are available. We utilise this observation in the following subsection, focusing on conjugacy of the Gaussian family with Gaussian likelihood.

## 3.3 Generalised squared family processes

We now specialise our choice of base measure and likelihood. Take the Radon-Nikodym derivative of the base measure $\mu$ with respect to Lebesgue measure to be a matrix Gaussian,

$$\frac{d\mu}{d\lambda}(\boldsymbol{\Omega}) \triangleq p(\boldsymbol{\Omega}) = \mathcal{N}(\boldsymbol{\omega} \mid \mathrm{vec}(\boldsymbol{M}), \boldsymbol{B} \otimes \boldsymbol{C}). \tag{3}$$

Take the train likelihood to be multivariate Gaussian with homoskedastic noise with variance $v^2$,

$$p(\boldsymbol{Y} \mid \boldsymbol{\Gamma}\boldsymbol{\Omega}^\top) = \mathcal{N}\big(\mathrm{vec}(\boldsymbol{Y}) \mid (\boldsymbol{I}_{d_2} \otimes \boldsymbol{\Gamma})\boldsymbol{\omega}, v^2 \boldsymbol{I}_{Nd_2}\big) \tag{4}$$

Take the test likelihood to be multivariate Gaussian with homoskedastic noise with variance $v'^2$,

$$p(\boldsymbol{Y}_* \mid \boldsymbol{\Gamma}_*\boldsymbol{\Omega}^\top) = \mathcal{N}\big(\mathrm{vec}(\boldsymbol{Y}_*) \mid (\boldsymbol{I}_{d_2} \otimes \boldsymbol{\Gamma}_*)\boldsymbol{\omega}, v'^2 \boldsymbol{I}_{N_*d_2}\big). \tag{5}$$

With a GSF prior, we refer to such a model as a *GSF process* (GSFP).

**Completing the square and parameter-integral factorisation**  By Proposition 4 and Gaussian conjugacy, in order to compute the posterior predictive distribution, we need only reverse the factorisation $\nu'(d\boldsymbol{\omega}) = p(\boldsymbol{Y}_* \mid \boldsymbol{\Gamma}_*\boldsymbol{\Omega}^\top)p(\boldsymbol{Y} \mid \boldsymbol{\Gamma}\boldsymbol{\Omega}^\top)\mu(d\boldsymbol{\omega})$ as $\nu'(d\boldsymbol{\omega}) = \mu(d\boldsymbol{\omega} \mid \boldsymbol{Y}, \boldsymbol{Y}_*)p(\boldsymbol{Y}_* \mid \boldsymbol{Y})p(\boldsymbol{Y})$, which is Gaussian, and same for $\nu(d\boldsymbol{\omega}) = p(\boldsymbol{Y} \mid \boldsymbol{\Gamma}\boldsymbol{\Omega}^\top)\mu(d\boldsymbol{\omega})$. The mean and covariance parameters of these Gaussian measures can be found using the same techniques as in vector-valued Bayesian linear regression. We show only the noise free posterior predictive distribution here, with more details in Appendix D.

**Corollary 6.** *Consider the special model* (3)*,* (4) *and* (5) *applied to Figure 3 (Right). Define* $\nu'(d\boldsymbol{\omega}) = p(\boldsymbol{Y}_* \mid \boldsymbol{\Gamma}_*\boldsymbol{\Omega}^\top)p(\boldsymbol{Y} \mid \boldsymbol{\Gamma}\boldsymbol{\Omega}^\top)\mu(\boldsymbol{\Omega})$ *and*

$$\boldsymbol{m}'_{\boldsymbol{Y}} = \mathrm{vec}(\boldsymbol{M}) + (\boldsymbol{B} \otimes \boldsymbol{C}\boldsymbol{\Gamma}^\top)(\boldsymbol{B} \otimes \boldsymbol{\Gamma}\boldsymbol{C}\boldsymbol{\Gamma}^\top + v^2 \boldsymbol{I}_{Nd_2})^{-1}\big(\mathrm{vec}(\boldsymbol{Y}) - (\boldsymbol{I}_{d_2} \otimes \boldsymbol{\Gamma})\mathrm{vec}(\boldsymbol{M})\big)$$

$$\boldsymbol{C}' = \boldsymbol{B} \otimes \boldsymbol{C} - (\boldsymbol{B} \otimes \boldsymbol{C}\boldsymbol{\Gamma}^\top)(\boldsymbol{B} \otimes \boldsymbol{\Gamma}\boldsymbol{C}\boldsymbol{\Gamma}^\top + v^2 \boldsymbol{I}_{Nd_2})^{-1}(\boldsymbol{B} \otimes \boldsymbol{\Gamma}\boldsymbol{C}).$$

*Suppose that $\boldsymbol{C}'$ and $\boldsymbol{\Gamma}_*$ are full-rank and $N_* \geq d_1$. Then the noise free predictive distribution* $P\big(d\boldsymbol{f}(\boldsymbol{X}_*) \mid \boldsymbol{Y}, \boldsymbol{\Gamma}, \boldsymbol{\Gamma}_*\big) = Q\big(d\boldsymbol{f}(\boldsymbol{X}_*) \mid \boldsymbol{M}, \nu'', \boldsymbol{\psi}'\big)$ *belongs to a GSF with modified feature mapping and modified base measure,*

$$\boldsymbol{\psi}'\big(\boldsymbol{f}(\boldsymbol{X}_*)\big) \triangleq \boldsymbol{\psi}\Big((\boldsymbol{I}_{d_2} \otimes \boldsymbol{\Gamma}_*)^\dagger \boldsymbol{f}(\boldsymbol{X}_*)\Big) = \boldsymbol{\psi}\Big(\mathrm{vec}\big(\boldsymbol{\Gamma}_*^\dagger \boldsymbol{f}(\boldsymbol{X}_*)\big)\Big)$$

$$\nu''\big(d\boldsymbol{f}(\boldsymbol{X}_*)\big) \triangleq \mathcal{N}\Big(\mathrm{vec}\big(\boldsymbol{f}(\boldsymbol{X}_*)\big) \mid (\boldsymbol{I}_{d_2} \otimes \boldsymbol{\Gamma}_*)\boldsymbol{m}'_{\boldsymbol{Y}}, (\boldsymbol{I}_{d_2} \otimes \boldsymbol{\Gamma}_*)\boldsymbol{C}'(\boldsymbol{I}_{d_2} \otimes \boldsymbol{\Gamma}_*^\top)\Big)d\boldsymbol{f}(\boldsymbol{X}_*). \tag{6}$$

Corollary 6 implies that under the Gaussian likelihood and base measure setting, the GSF $\mathbb{Q}_{\boldsymbol{M},\mathbb{A},\Psi}$ is a conjugate class, where $\Psi$ is the class of transformations obtained by composition of $\boldsymbol{\psi}$ with linear transformations and $\mathbb{A}$ involves deformation via Gaussian conjugate updates, as per (6).

## 3.4 Related work

To the best of our knowledge, we are the first to consider squared families as priors and posteriors. The nature of the results concerning likelihoods is fundamentally different to that of priors and posteriors, because for example, one does not attempt to estimate parameters in forming a posterior, one merely attempts to form a posterior belief. Nevertheless, existing work on special cases of squared family likelihoods is relevant in that expressive power, flexibility and tractability of densities is intuitively relevant to both likelihoods and posteriors, and uncertainty quantification more generally.

**Kernel methods**  Building off the general results of modelling non-negative functions using squared elements of RKHS [Marteau-Ferey et al., 2020], Rudi and Ciliberto [2021] model probability distributions, with tractable marginalisation and closure under multiplication. They fit likelihood models within this class by minimising a regularised L2 distance between the target likelihood and the model. While these models are nonparametric in construction, estimates resulting from minimising the objective satisfy a representer theorem type instantiation in finite dimensions. The authors show that the model admits both favourable representational capability (in being able to represent a $\beta$-times differentiable target density with error less than with number of parameters scaling like $\mathcal{O}(\epsilon^{-r/\beta}(\log(1/\epsilon))^{r/2})$, where $r$ is the dimension of the domain of the distribution), as well as statistical estimation properties (as likelihoods, the proposed estimate converges at a rate of $N^{-\frac{\beta}{2\beta+r}}(\log N)^{r/2}$, where $N$ is the number of datapoints).

**Probabilistic circuits** Probabilistic circuits [Choi et al., 2020] provide a computational framework for tractable probabilistic modelling. They are constructed as graphs of connected units, and by constraining the graph, allow for tractable marginalisation. However, they need to impose some structure on the units or the connections in order to ensure nonnegativity. Squared probabilistic circuits [Sladek, 2023, Loconte et al., 2023a,b, Loconte and Vergari, 2025] bypass this constraint by squaring the output of the circuit. Sums of squared circuits can also be instantiated [Loconte et al., 2024]. The focus on such works is typically in showing tractable (polynomial or indeed quadratic time/memory in the number of units) computation of normalising constants or marginalisation, as well as the inclusions of the function spaces imposed by different classes of probabilistic circuits.

**Neural networks and finite feature models** Previous work [Tsuchida et al., 2025] has modelled likelihoods as proportional to the squared Euclidean norm of a function of the form $\Theta\psi(x)$, where $\Theta$ is a matrix and $\psi$ is a vector-valued function. Provided the features $\psi$ are rich enough, such models admit a universal approximation property, typically approximating the squared L2 distance between a square root of a target density and the model at a rate of $\mathcal{O}(1/n)$, where $n$ is a parameter count. As likelihoods, they learn target densities with a KL divergence to the target density decreasing at a rate of $\mathcal{O}(1/\sqrt{N})$, where $N$ is the number of samples. Such families of models have tractable normalising constants, Fisher information and statistical divergences. Furthermore, in some special cases, such families have not only tractable but indeed closed-form normalising constants, Fisher information and statistical divergences. An example studied previously was squared neural families [Tsuchida et al., 2023], where is a single hidden layer neural network.

**Applications** Poisson point processes (PPPs) are helpful variations of density modelling, where one models an intensity instead of a density. Whereas in density modelling, realisations are i.i.d. draws from a target density, in intensity modelling, realisations are conditionally i.i.d. draws from a target density given the number of realisations. The number of realisations follows a Poisson distribution with an expected number of points equal to the integral of the intensity over the domain. Flaxman et al. [2017] used (frequentist) squared elements of RKHSs to model intensity functions, whereas Walder and Bishop [2017] used a (Bayesian) squared Gaussian process prior to model intensity functions. We note that placing a prior over a function which when squared gives a intensity/density (likelihood) is completely disjoint to the problem we are solving here, which is to use a squared function as a prior density. Probabilistic circuits have been applied in converting knowledge graph embeddings into generative models [Loconte et al., 2023a]. Squared neural family likelihoods have been applied to label distribution learning [Zhang et al., 2025], which is a kind of variation on classical regression where instead of the target label being a single class, the target variable is a discrete distribution (i.e. an element of the simplex). Hence, here squared neural family models model a distribution over distributions with a closed-form normalising constant, expectation, variance, and covariance conditioned on input samples. Probabilistic modelling is leveraged to provide confidence intervals, conformal predictions, active learning, and model ensembling.

## 4 Experiments on few-shot learning

**Few shot learning setting** The hyperparameters of a nonparametric probabilistic model (such as a GP or GSFP) can be adapted to a single training dataset $(X, Y)$ with $N$ examples, by maximising the marginal likelihood $p(Y \mid X)$ under the model with respect to the model hyperparameters, also referred to as type II maximum marginal likelihood [Rasmussen and Williams, 2006, § 5.4]. It may also happen that a meta dataset, consisting of $J$ datasets $\{(X_j, Y_j)\}_{j=1}^{J}$, is assumed to arise as $J$ samples from a model sharing the same hyperparameters, in which case the marginal likelihood $p(\{Y_j\}_{j=1}^{J} \mid \{X_j\}_{j=1}^{J})$ still serves as a natural objective for tuning model hyperparameters [Rasmussen and Williams, 2006, p. 115]. In this case, each of the $(X_j, Y_j)$ consists of $N_j$ examples, where $N_j$ may be constant $N$ across all $J$ datasets, but not necessarily. At inference time, when a new *support dataset* $(X_1^*, Y_1^*)$ and a new set of *query inputs* $X_2^*$ are given, one may form the posterior predictive $p(Y_2^* \mid X_2^*, X_1^*, Y_1^*)$ using the hyperparameters obtained by maximising the marginal likelihood of the tasks. This classical setting arises from point estimation of the hyperparameters under the assumption of shared point hyperparameters (see right side of Figure 3), and has traditionally been referred to under the umbrella of multi-task learning [Caruana, 1997, Minka and Picard, 1997].

This setting has recently received renewed attention under the name of *few-shot learning*, and GP models have been empirically shown to outperform other more recent models on modern benchmarks,

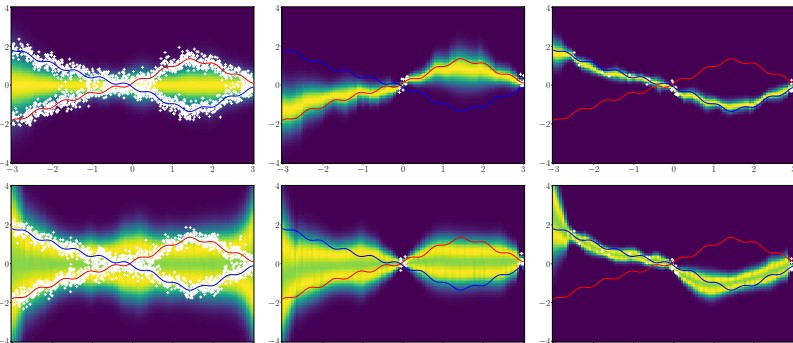

Figure 4: When can a multimodal prior be helpful? (Top) GP using deep learning features (deep kernel learning [Wilson et al., 2016]). (Bottom) GSFP using deep learning features. (Left) Prior predictive, trained using points (white) drawn from a multimodal distribution over functions (with mean given by red and blue). The GFSP is bimodal, whereas the GP is always unimodal. (Middle) Posteriors for both models given some ambiguous points, which may belong to either red or blue modes. The GP incorrectly places all of its belief over one of the modes, whereas the GFSP reserves belief for both modes. (Right) Given additional data from $x \leq 0$, both identify the correct mode.

both in the regression and classification setting [Patacchiola et al., 2020] (Deep kernel transfer, or DKT). In the regression setting, further enhancements to GP models, which use continuous normalising flows to transform GP marginal predictive posterior distributions into more complicated distributions, have further been used to improve upon GPs [Sendera et al., 2021] (Non-Gaussian Gaussian processes, or NGGP). Such enhancements address GP prior limitations; namely lack of an ability to probabilistically model skewed, non-Gaussian-tailed or multimodal beliefs. Our GSFP also offer a means to address such limitations (see Figure 4), and are therefore an excellent candidate for few-shot learning. We consider the same scalar-valued ($d_2 = 1$) regression benchmarks as Sendera et al. [2021], who in turn add to the regression benchmarks of Patacchiola et al. [2020].

**Implementation details**   We learn a prior predictive distribution, which is further broken into a prior parameter distribution belonging to a GSF $\mathbb{Q}_{\mathbb{M}, \mathbb{A}, \Psi}$ and a (deterministic) feature extractor $\gamma$, by attempting to maximise the marginal likelihood (see Proposition 2) with respect to some tuneable decision variables. The decision variables are an unconstrained $\Theta$, with $\Theta^\top \Theta \in \mathbb{M}$ and a feature mapping $\psi$ (which together define the prior over $\Omega$ in (2)), as well as the tuneable parameters of the neural network $\gamma$. We further parameterise the feature mapping $\psi$ as the hidden layer of a learnable neural network, $\psi(\omega) = \sigma(W\omega + b)$, with decision variables $W$ and $b$. Updates within the set $\mathbb{A}$ happen automatically as part of the Bayesian inference pipeline and are not part of the decision variables in the marginal likelihood (see Proposition 2). We reimplemented competitor models DKT [Patacchiola et al., 2020] and NGGP [Sendera et al., 2021], as we were not able to get the originally supplied code to run, and through extensive hyperparameter tuning were able to obtain results better than what they originally reported in all but two datasets (NDX100 and EEG). Full hyperparameter details and run times are given in Appendix F. Results are shown in Table 4. We observe that GFSP obtains the lowest NLL on 11 out of 12 settings, although sometimes is well within one standard deviation of the best competing method. While our method is designed for situations in which a multimodal belief is helpful, e.g. few-shot learning, we can also apply it to vanilla regression. See Appendix G.

## 5   Discussion, limitations and conclusion

**Limitations**   The main computational hurdle in working with GSFs is computing the (exact) normalising constant via the squared family kernel, which has a time complexity of $\mathcal{O}(n^2 d)$, where $n$ is the size of the hidden layer and $d$ is the dimension of the random variable. In practice, the training time of GSFP is slightly longer than that of DKT, but substantially less than that of NGGP (see Table F.9). The inference time is much lower than NGGP, but slightly more than DKT. As with competitors DKT and NGGP, which are also based on (potentially finite-feature) GP methods, our

Table 1: Benchmark results showing test NLLs for regression tasks across 6 meta datasets. For each meta dataset, we train 5 meta models with different random seed via maximum marginal likelihood over the $J$ datasets $\{(\boldsymbol{X}_j, \boldsymbol{Y}_j)\}_{j=1}^J$. At testing time, we condition (train) on a support (few shot) dataset $(\boldsymbol{X}_1^*, \boldsymbol{Y}_1^*)$ and evaluate the test NLL on the query dataset $(\boldsymbol{X}_2^*, \boldsymbol{Y}_2^*)$. We evaluate the test NLL on these few shot datasets for 500 random shuffles of the test meta dataset. In total, $5 \times 500 = 2{,}500$ models are trained. Each entry shows the mean $\pm$ standard deviation over $2{,}500$ model evaluations. Each testing scenario includes in-range and out-of-range, where the support and query set is either similar to or different to the $J$ training datasets. Best results in **bold**, second best underlined.

| Methods | | Sines[Patacchiola et al., 2020] | | Mixed-Noise Sines [Sendera et al., 2021] | | QMUL [Gong et al., 1996] | |
|---|---|---|---|---|---|---|---|
| Name | Kernel | $\mathrm{NLL_{In}}\downarrow$ | $\mathrm{NLL_{Out}}\downarrow$ | $\mathrm{NLL_{In}}\downarrow$ | $\mathrm{NLL_{Out}}\downarrow$ | $\mathrm{NLL_{In}}\downarrow$ | $\mathrm{NLL_{Out}}\downarrow$ |
| DKT | RBF | $-0.71\pm0.06$ | $-0.62\pm0.08$ | $0.76\pm0.06$ | $\underline{1.07\pm0.13}$ | $-0.73\pm0.19$ | $-0.69\pm0.21$ |
| | Spectral | $-0.82\pm0.05$ | $\underline{-0.79\pm0.06}$ | $0.38\pm0.17$ | $1.78\pm0.49$ | $-0.67\pm0.24$ | $-0.64\pm0.20$ |
| | NN Linear | $-0.76\pm0.09$ | $0.30\pm1.07$ | $0.47\pm0.25$ | $2.31\pm0.72$ | $-0.66\pm0.49$ | $0.87\pm1.18$ |
| NGGP | RBF | $-0.74\pm0.06$ | $-0.56\pm0.14$ | $\underline{0.28\pm0.13}$ | $2.45\pm0.87$ | $-0.40\pm0.69$ | $0.15\pm0.70$ |
| | Spectral | $\underline{-0.84\pm0.05}$ | $-0.73\pm0.07$ | $0.38\pm0.17$ | $1.78\pm0.49$ | $\underline{-0.99\pm0.31}$ | $\underline{-0.77\pm0.31}$ |
| | NN Linear | $-0.80\pm0.06$ | $0.27\pm1.26$ | $\mathbf{0.24\pm0.89}$ | $2.44\pm1.32$ | $0.04\pm1.34$ | $1.41\pm1.27$ |
| GSFP | NN Linear | $\mathbf{-0.85\pm0.06}$ | $\mathbf{-0.83\pm0.07}$ | $0.46\pm0.10$ | $\mathbf{0.98\pm0.21}$ | $\mathbf{-1.09\pm0.12}$ | $\mathbf{-1.15\pm0.14}$ |
| Methods | | NDX100 [Qin et al., 2017] | | EEG [Fernandez-Fraga et al., 2019] | | Power [Hebrail and Berard, 2006] | |
| Name | Kernel | $\mathrm{NLL_{In}}\downarrow$ | $\mathrm{NLL_{Out}}\downarrow$ | $\mathrm{NLL_{In}}\downarrow$ | $\mathrm{NLL_{Out}}\downarrow$ | $\mathrm{NLL_{In}}\downarrow$ | $\mathrm{NLL_{Out}}\downarrow$ |
| DKT | RBF | $-1.16\pm0.03$ | $-1.17\pm0.03$ | $-1.10\pm0.54$ | $-1.16\pm1.09$ | $\underline{-0.47\pm0.62}$ | $-0.34\pm0.70$ |
| | Spectral | $-0.46\pm0.43$ | $-0.63\pm0.30$ | $\underline{-1.32\pm0.31}$ | $0.84\pm0.67$ | $\underline{-0.47\pm0.62}$ | $-0.35\pm0.70$ |
| | NN Linear | $\underline{-2.16\pm1.61}$ | $\underline{-2.41\pm0.96}$ | $-1.10\pm0.52$ | $-1.02\pm1.06$ | $-0.42\pm0.68$ | $-0.33\pm0.73$ |
| NGGP | RBF | $1.16\pm0.01$ | $1.15\pm0.01$ | $-0.96\pm0.47$ | $0.05\pm0.32$ | $1.15\pm0.02$ | $1.15\pm0.02$ |
| | Spectral | $-1.32\pm0.31$ | $0.84\pm0.67$ | $-1.07\pm0.30$ | $-0.19\pm0.37$ | $1.35\pm0.01$ | $1.35\pm0.01$ |
| | NN Linear | $-2.07\pm1.26$ | $-2.32\pm0.62$ | $-1.05\pm0.59$ | $\underline{-1.18\pm0.89}$ | $-0.27\pm0.40$ | $-0.27\pm0.40$ |
| GSFP | NN Linear | $\mathbf{-2.49\pm1.28}$ | $\mathbf{-2.75\pm0.53}$ | $\mathbf{-1.49\pm0.44}$ | $\mathbf{-1.38\pm0.81}$ | $\mathbf{-0.71\pm0.39}$ | $\mathbf{-0.65\pm0.45}$ |

method has $\mathcal{O}(N^2)$ and $\mathcal{O}(N^3)$ memory and time complexity for inference (or $\mathcal{O}(nN)$ and $\mathcal{O}(n^3)$ for finite-feature models), where $N$ is the number of training points.

**Computational complexity of Bayesian inference**  In the general case, Bayesian inference is computationally difficult, unless some heavy restriction is placed on the densities (e.g. log concave). This is especially difficult when the dimension is large, due to the curse of dimensionality and the exponential growth of volume. In practice, if one desires an exact computation, one needs to restrict the class of probability distributions. When we restrict the class to GSFs, we are able to compute exact posteriors. The complexity of this calculation is governed by the inner product of the parameter matrix $\boldsymbol{M}$ and the kernel matrix $\boldsymbol{K}_{\mu,\psi}$ (as in (1)). Assuming the kernel matrix is known, this inner product has complexity $\mathcal{O}(n^2)$, where $n$ is the number of parameters in the GSF. Crucially, dependence of the complexity on $d$ only appears through the calculation of the kernel matrix $\boldsymbol{K}_{\mu,\psi}$. In practice, the kernel matrices $\boldsymbol{K}_{\mu,\psi}$ can be computed exactly in linear time in $d$. This leads to a combined complexity of $\mathcal{O}(dn^2)$, i.e. exact inference linear in dimension $d$, which is a huge improvement over inexact inference exponential in dimension $d$. This improvement comes with the restriction of the class of densities to GSFs, however this restriction is not too severe because GSFs are rich density approximators. More precisely navigating the trade-off between universal approximation (making $n$ large) and computational efficiency (making $n$ small) is an important direction for future work.

**Conclusion**  GSF likelihoods appear in disguise in a wide range of applications, including Poisson point processes [Flaxman et al., 2017, Walder and Bishop, 2017, Tsuchida et al., 2024] generative models [Loconte et al., 2023a], and most fundamentally, density estimation [Rudi and Ciliberto, 2021]. Here we considered the orthogonal setting of GSF priors. Using the parameter-integral factorisation, we found that GSFs form conjugate priors and that for many likelihoods of interest, admit closed-form Bayesian updates. Such a closed-form update allows one to generalise Gaussian parameter prior and Gaussian likelihood regression in feature space models to GSF priors, allowing for multimodal and rich expressions of uncertainty. We empirically demonstrated that GSFPs perform better than or as well as other models on a range of benchmark few-shot regression problems. In future, it should be possible to show asymptotic normality of maximum marginal likelihood estimates, as well as provide generalisation bounds for the problem of density estimation under a hierarchical model, by using the fact (see Footnote 2) that the marginal likelihood asymptotically belongs to a GSF and using known results for GSF likelihoods.

## Acknowledgements

DS was partially supported by the Responsible AI Research Centre (RAIR).

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

# A    Conjugacy of exponential families

## A.1    General formulation

Here we review the well-known conjugacy of exponential families. Let $\boldsymbol{t} : \mathbb{X} \to \mathbb{R}^{a_2}$ with $\mathbb{X} \subseteq \mathbb{R}^{a_1}$, and let $H$ be a nonnegative measure. Let $A$ be such that

$$P(d\boldsymbol{x} \mid \boldsymbol{\eta}) = \exp\left(\boldsymbol{\eta}^\top \boldsymbol{t}(\boldsymbol{x}) - A(\boldsymbol{\eta})\right) H(d\boldsymbol{x})$$

integrates to 1. The set $\{P(d\boldsymbol{x} \mid \boldsymbol{\eta})\}_{\boldsymbol{\eta} \in \{\boldsymbol{\eta}' \mid 0 < \int \exp(\boldsymbol{\eta}'^\top \boldsymbol{t}(\boldsymbol{x})) H(d\boldsymbol{x}) < \infty\}}$ is called an exponential family. A conjugate family for an exponential family likelihood has densities of the form

$$p(\boldsymbol{\eta} \mid \boldsymbol{\chi}, r) = \frac{\exp\left(\boldsymbol{\chi}^\top \boldsymbol{\eta} - r A(\boldsymbol{\eta})\right)}{\int \exp\left(\boldsymbol{\chi}^\top \boldsymbol{\eta} - r A(\boldsymbol{\eta})\right) d\boldsymbol{\eta}} \propto \exp\left(-r A(\boldsymbol{\eta})\right) \exp(\boldsymbol{\chi}^\top \boldsymbol{\eta})$$

That is, a conjugate family is $\{p(\boldsymbol{\eta} \mid \boldsymbol{\chi}, r)\}_{\boldsymbol{\chi} \in \mathbb{R}^{a_2}, r > 0}$. Note that the right expression is in the form of an exponential family with canonical parameters $\widetilde{\boldsymbol{\eta}} = (\boldsymbol{\chi}, r)$ and sufficient statistic $\widetilde{\boldsymbol{t}}(\boldsymbol{\eta}) = (\boldsymbol{\eta}, -A(\boldsymbol{\eta}))$. The conjugate prior has the exponential family form

$$p(\boldsymbol{\eta} \mid \widetilde{\boldsymbol{\eta}}) = \exp\left(\widetilde{\boldsymbol{\eta}}^\top \widetilde{\boldsymbol{t}}(\boldsymbol{\eta}) - \widetilde{A}(\widetilde{\boldsymbol{\eta}})\right), \qquad \widetilde{A}(\widetilde{\boldsymbol{\eta}}) = \log \int \exp\left(\widetilde{\boldsymbol{\eta}}^\top \widetilde{\boldsymbol{t}}(\boldsymbol{\eta})\right) d\boldsymbol{\eta}.$$

Given iid observations $\boldsymbol{X} \in \mathbb{R}^{N \times a_1}$ from the likelihood, the posterior is obtained by updating the parameters of the prior with the sufficient statistics,

$$p(\boldsymbol{\eta} \mid \boldsymbol{X}, \boldsymbol{\chi}, r) = p\left(\boldsymbol{\eta} \Big| \boldsymbol{\chi} + \sum_{i=1}^{N} \boldsymbol{t}(\boldsymbol{x}_i), r + N\right).$$

## A.2    Examples

**Poisson-Gamma**    Take $\mathbb{X} = \{0, 1, \ldots\}$, $h(x) = \frac{1}{x}$ and $t(x) = x$. This results in a Poisson likelihood, with rate $\lambda = \exp \eta$ and $A(\eta) = \exp(\eta)$. The conjugate family is a set of distributions of the form

$$p(\eta \mid \chi, r) \propto \exp\left(A(-r\eta)\right) \exp(\chi \eta) = \exp\left(-r \exp(\eta)\right) \exp(\chi \eta).$$

Taking $\lambda = \exp \eta$, and accounting for the Jacobian of the transformation, we identify the family of densities of the form

$$p(\lambda \mid \chi, r) \propto \lambda^{\chi - 1} \exp(-r\lambda),$$

which are Gamma densities with rate $\lambda$ and shape $\chi$. These have a known (divisive) normalising constant given by $\Gamma(\chi)/r^\chi$, where here $\Gamma$ denotes the Gamma function. The posterior density is then

$$p(\lambda \mid \boldsymbol{X}, \chi, r) \propto \lambda^{\chi + \sum_{i=1}^{N} t(\boldsymbol{x}_i) - 1} \exp(-(r + N)\lambda),$$

with known normalising constant. The corresponding prior distribution over $\eta = \log \lambda$, as an exponential family, is the exponential-gamma distribution, with

$$p(\eta \mid \boldsymbol{X}, \chi, r) = \frac{r^\chi}{\Gamma(\chi)} \exp\left(\chi \eta\right) \exp(-r \exp(\eta)).$$

An example of Poisson-Gamma updates is shown in Figure 5.

**Gaussian-Gaussian**    Take $\mathbb{X} = \mathbb{R}$, $h(x) = \exp\left(-x^2/(2s^2)\right) (2\pi s^2)^{-1/2}$, and $t(x) = x/s$ for some known standard deviation $s > 0$. This results in a Gaussian likelihood, with known variance $s^2$, mean $s\eta$ and $A(\eta) = \eta^2/2$. The conjugate family is a set of distributions of the form

$$p(\eta \mid \chi, r) \propto \exp(-r\eta^2/2) \exp(\chi \eta),$$

which are Gaussian densities with variance $1/r$ and mean $\chi/r$. These have a known (divisive) normalising constant, and

$$p(\eta \mid \chi, r) = \frac{1}{\sqrt{2\pi r^{-1}}} \exp\left(-\frac{1}{2r^{-1}}(\eta - \chi/r)^2\right).$$

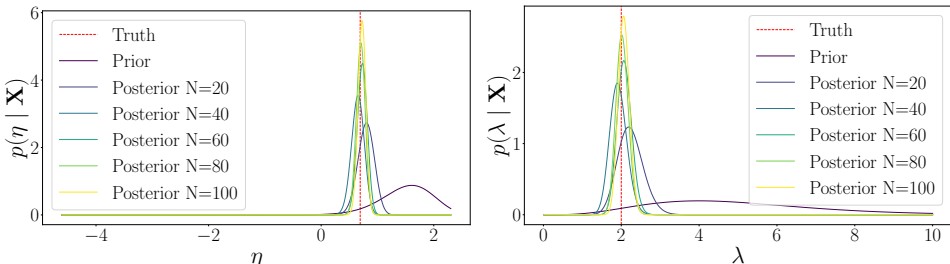

Figure 5: Bayesian inference in a well-specified Poisson model, using a Gamma conjugate family for rate $\lambda$ (or, in canonical parameterisation, an exponential-Gamma conjugate family for $\eta = \log \lambda$). Shown are the posterior distributions for the parameter given $N = 0, 20 \ldots, 100$ samples from the likelihood.

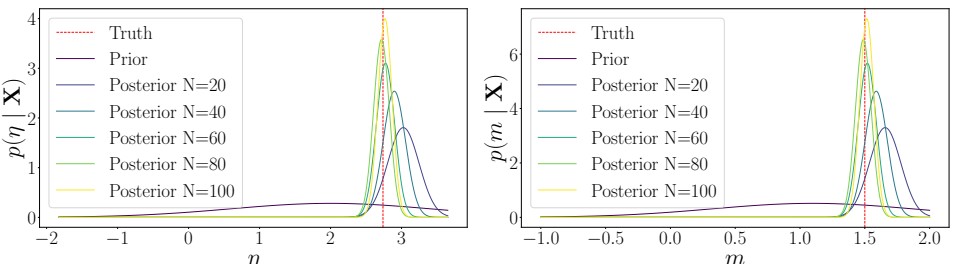

Figure 6: Bayesian inference in a well-specified Gaussian model (with known variance), using a Gaussian conjugate family for mean $m$ (in canonical parameterisation, the conjugate family is also Gaussian). Shown are the posterior distributions for the parameter given $N = 0, 20, \ldots, 100$ samples from the likelihood.

Under the change of variables $m = s\eta$ for mean $m$,

$$p(m \mid \chi, r) = \mathcal{N}(\chi s/r, s^2/r) \qquad \text{and}$$

$$p(m \mid \boldsymbol{X}, \chi, r) = \mathcal{N}\left( (\chi + \sum_{i=1}^{N} x_i)s/(r + N), s^2/(r + N) \right).$$

An example of Gaussian-Gaussian updates is shown in Figure 6.

## B Examples of squared family kernels

### B.1 Exponential family base measures and exponential features

Take the base measure $\mu$ to belong to any exponential family with sufficient statistic $\widetilde{t}$. Take the feature $\boldsymbol{\psi}$ to be $\boldsymbol{\psi}(\boldsymbol{\omega}) = \exp\left(\boldsymbol{W}\widetilde{\boldsymbol{t}}(\boldsymbol{\omega})/2\right)$. Then the $ij$th entry of the squared family kernel $\boldsymbol{K}_{\mu,\boldsymbol{\psi}}$ is

$$(\boldsymbol{K}_{\mu,\boldsymbol{\psi}})_{ij} = \int_{\Omega} \exp\left((\boldsymbol{w}_i + \boldsymbol{w}_j)^{\top}\widetilde{\boldsymbol{t}}(\boldsymbol{\omega})/2\right) \mu(d\boldsymbol{\omega}),$$

where $\boldsymbol{w}_i$ is the $i$th row of $\boldsymbol{W}$. This is simply the moment generating function of the sufficient statistic $\widetilde{t}$ under the exponential family, which is available in closed-form in terms of the log normalising constant $\widetilde{A}(\widetilde{\eta})$ of the exponential family,

$$(\boldsymbol{K}_{\mu,\boldsymbol{\psi}})_{ij} = \exp\left(\widetilde{A}\left(\widetilde{\boldsymbol{\eta}} + (\boldsymbol{w}_i + \boldsymbol{w}_j)/2\right) - \widetilde{A}(\widetilde{\boldsymbol{\eta}})\right),$$

where $\boldsymbol{\eta}$ is the canonical parameter of the exponential family. Recall Remark 5, that if $\mu$ itself belongs to a conjugate prior family for the likelihood $p(\boldsymbol{U} \mid \cdot)$, then $\nu(\cdot) = \mu(\cdot)p(\boldsymbol{U} \mid \cdot)$ remains within the conjugate family, up to a multiplication by the marginal likelihood, and therefore the squared family kernel $\boldsymbol{K}_{\nu,\boldsymbol{\psi}}$ is available in closed-form for all posterior updates. By Appendix A, $\mu$ does indeed belong to a conjugate family. Log normalising constants are available for many exponential families which are known to be conjugate priors for exponential family likelihoods [Diaconis and Ylvisaker, 1979] (see e.g. Appendix A and closed-form normalising constants in Nielsen and Garcia [2024]).

### B.2 Normalised base measures and random Fourier features

Bochner's theorem says that every stationary kernel is the Fourier transform of a nonnegative probability measure, and vice versa [Rahimi and Recht, 2007, for example]. Hence for real-valued stationary squared family kernels and probability base measures $\mu$,

$$
\begin{aligned}
(\boldsymbol{K}_{\mu,\boldsymbol{\psi}})_{ij} &= \int_{\Omega} \exp\left(\mathrm{i}(\boldsymbol{w}_i - \boldsymbol{w}_j)^{\top}\boldsymbol{\omega}\right) \mu(d\boldsymbol{\omega}) \\
&= \int_{\Omega} \psi_i(\boldsymbol{\omega})\psi_j(\boldsymbol{\omega})\mu(d\boldsymbol{\omega}),
\end{aligned}
$$

where $\psi_i(\boldsymbol{\omega}) = \cos(\boldsymbol{w}_i^{\top}\boldsymbol{\omega}) + \mathrm{i}\sin(\boldsymbol{w}_i^{\top}\boldsymbol{\omega})$ and $\mathrm{i}$ is the imaginary unit. This allows the use of base measures with cosine and sine features, provided the base measure has a closed-form Fourier transform. For example, the Gaussian, Laplace and Cauchy base measures all have closed-form kernels (corresponding with Gaussian, Cauchy and Laplacian kernels respectively). Real-valued cosine and sine transforms are also available for many densities, including the Gaussian density.

### B.3 Gaussian family base measures and neural network features

Integrals of the form of the squared family kernel arise as so-called neural network Gaussian process kernels, when the features are hidden layers of neural networks and the base measure is Gaussian, That is, the features take the form $\boldsymbol{\psi}(\boldsymbol{\omega}) = \sigma(\boldsymbol{W}\boldsymbol{\omega} + \boldsymbol{b})$ for some weight and bias parameters $\boldsymbol{W}$ and $\boldsymbol{b}$, and some activation function $\sigma$, and

$$(\boldsymbol{K}_{\mu,\boldsymbol{\psi}})_{ij} = \mathbb{E}_{\boldsymbol{\omega}\sim\mathcal{N}}\left[\sigma(\boldsymbol{w}_i^{\top}\boldsymbol{\omega} + b_i)\sigma(\boldsymbol{w}_j^{\top}\boldsymbol{\omega} + b_j)\right].$$

Example activation functions with known closed form include the error function [Williams, 1996], the ReLU (and other rectified monomial) function [Cho and Saul, 2009] and sine and cosines [Pearce et al., 2020]. Further examples are given in Table 1 of Tsuchida et al. [2023] and Table 1 of Han et al. [2022], including monomials, GeLU, snake and approximations to well-behaved functions.

### B.4 Table of some closed-form squared families

| Domain $\Omega$ | Base measure $\mu$ | Feature $\boldsymbol{\psi}(\boldsymbol{\omega})$ | Reference/Notes |
|---|---|---|---|
| $\mathbb{R}^d$ | $\mathcal{N}(\mathbf{0}, \mathrm{Cov})$ | $\mathrm{erf}(\boldsymbol{W}\boldsymbol{\omega})$ | Williams [1996] |
| $\mathbb{R}^d$ | $\mathcal{N}(\mathbf{0}, \mathrm{Cov})$ | $\mathrm{ReLU}(\boldsymbol{W}\boldsymbol{\omega})$ | Cho and Saul [2009] |
| $\mathbb{R}^d$ | $\mathcal{N}(\mathbf{0}, \mathrm{Cov})$ | $\mathrm{GELU}(\boldsymbol{W}\boldsymbol{\omega})$ | Tsuchida et al. [2021] |
| $\mathbb{R}^d$ | $\mathcal{N}(\mathrm{mean}, \mathrm{Cov})$ | $\mathrm{Snake}(\boldsymbol{W}\boldsymbol{\omega} + \boldsymbol{b})$ | Tsuchida et al. [2023] |
| $\mathbb{R}^d$ | $\mathcal{N}(\mathrm{mean}, \mathrm{Cov})$ | $\exp(\boldsymbol{W}\boldsymbol{\omega} + \boldsymbol{b})$ | Nielsen and Garcia [2024] |
| $\{0, 1, 2, \ldots\}$ | $(\omega!)^{-1}\nu$ | $\exp(\boldsymbol{W}\boldsymbol{\omega} + \boldsymbol{b})$ | Nielsen and Garcia [2024] |
| $\mathbb{S}^{d-1}$ | Uniform | $\exp(\boldsymbol{W}\boldsymbol{\omega} + \boldsymbol{b})$ | Brown [1986] |

Table 2: All of these settings admit closed-form squared family kernels $\boldsymbol{K}_{\mu,\psi}$. In each case, the closed-form kernel is computed in linear time in the dimension $d$. Here erf denotes the error function, $\mathbb{S}^{d-1}$ denotes the unit hypersphere, Snake denotes the snake activation function [Ziyin et al., 2020]. Even further examples are given in Han et al. [2022, table 1] and Tsuchida et al. [2023, table 1].

## C   Proofs

**Proposition 3.** *Consider the model in Figure 3 (Left). Define a measure $\nu(d\boldsymbol{\omega}) = p(\boldsymbol{U} \mid \boldsymbol{\omega})\mu(d\boldsymbol{\omega})$. The posterior $P(d\boldsymbol{\omega} \mid \boldsymbol{U}) = Q(d\boldsymbol{\omega} \mid \boldsymbol{M}, \nu, \boldsymbol{\psi})$ belongs to a GSF with base measure $\nu$, feature mapping $\boldsymbol{\psi}$ and parameter $\boldsymbol{M}$.*

*Proof.* The posterior is the product of the likelihood and prior divided by the marginal likelihood,

$$\frac{1}{\text{Tr}\left(\boldsymbol{M}\boldsymbol{K}_{\mu,\psi}\right)p(\boldsymbol{U})} \langle \boldsymbol{M}\boldsymbol{\psi}(\boldsymbol{\omega}), \boldsymbol{\psi}(\boldsymbol{\omega})\rangle p(\boldsymbol{U} \mid \boldsymbol{\omega})\mu(d\boldsymbol{\omega})$$
$$\propto \langle \boldsymbol{M}\boldsymbol{\psi}(\boldsymbol{\omega}), \boldsymbol{\psi}(\boldsymbol{\omega})\rangle p(\boldsymbol{U} \mid \boldsymbol{\omega})\mu(d\boldsymbol{\omega})$$
$$\propto Q(d\boldsymbol{\omega} \mid \boldsymbol{M}, \nu, \boldsymbol{\psi}).$$

$\square$

**Proposition 4.** *Consider the model in Figure 3 (Left). Define a measure $\nu(d\boldsymbol{\omega}) = p(\boldsymbol{U} \mid \boldsymbol{\omega})\mu(d\boldsymbol{\omega})$ and $\nu'(d\boldsymbol{\omega}) = p(\boldsymbol{U}_* \mid \boldsymbol{\omega})\nu(d\boldsymbol{\omega})$, where, abusing notation, the train likelihood $p(\boldsymbol{U} \mid \boldsymbol{\omega})$ and test likelihood $p(\boldsymbol{U}_* \mid \boldsymbol{\omega})$ are not necessarily the same. The posterior predictive is the ratio of normalising constants, $p(\boldsymbol{U}_* \mid \boldsymbol{U}) = \text{Tr}\left(\boldsymbol{M}\boldsymbol{K}_{\nu',\psi}\right)/\text{Tr}\left(\boldsymbol{M}\boldsymbol{K}_{\nu,\psi}\right)$.*

*Proof.* This follows by applying Proposition 2 twice, or directly. Directly, the posterior predictive is obtained by marginalising out the predictive distribution given the parameters with respect to the parameter posterior. That is,

$$p(\boldsymbol{U}_* \mid \boldsymbol{U}) = \int_{\Omega} p(\boldsymbol{U}_* \mid \boldsymbol{\omega})P(d\boldsymbol{\omega} \mid \boldsymbol{u})$$

$$= \int_{\Omega} p(\boldsymbol{U}_* \mid \boldsymbol{\omega})Q(d\boldsymbol{\omega} \mid \boldsymbol{M}, \nu\boldsymbol{\psi})$$

$$= \frac{1}{\text{Tr}\left(\boldsymbol{M}\boldsymbol{K}_{\nu,\psi}\right)} \int_{\Omega} \text{Tr}\left(\boldsymbol{M}\boldsymbol{\psi}(\boldsymbol{\omega})\boldsymbol{\psi}(\boldsymbol{\omega})^{\top}\right)p(\boldsymbol{U}_* \mid \boldsymbol{\omega})\nu(d\boldsymbol{\omega})$$

$$= \frac{\text{Tr}\left(\boldsymbol{M}\boldsymbol{K}_{\nu',\psi}\right)}{\text{Tr}\left(\boldsymbol{M}\boldsymbol{K}_{\nu,\psi}\right)}.$$

$\square$

## D   Squared probability process calculations

The derivation of the following two corollaries may appear long and symbol-laden but is essentially just mixing the parameter-integral factorisation (see § 2) with the standard "completing the square" technique in Gaussian linear regression [Bishop, 2006, §2.3.1] [Rasmussen and Williams, 2006, §2.1.1].

**Corollary 7.** *Consider the special model (3), (4) and (5) applied to Figure 3 (Right). Define $\nu'(d\boldsymbol{\omega}) = p(\boldsymbol{Y}_* \mid \boldsymbol{\Gamma}_*\boldsymbol{\Omega}^{\top})p(\boldsymbol{Y} \mid \boldsymbol{\Gamma}\boldsymbol{\Omega}^{\top})\mu(\boldsymbol{\Omega})$. The Radon-Nikodym derivatives of $\nu$ and $\nu'$ are the same as in the finite-feature Gaussian process regression setting. That is,*

$$\frac{d\nu}{d\lambda}(d\boldsymbol{\omega}) = \mathcal{N}\left(\boldsymbol{\omega} \mid \boldsymbol{m}'_{\boldsymbol{Y}}, \boldsymbol{C}'\right)\mathcal{N}_{\boldsymbol{Y}}, \quad \text{where}$$

$$\boldsymbol{m}'_{\boldsymbol{Y}} = \text{vec}(\boldsymbol{M}) + (\boldsymbol{B} \otimes \boldsymbol{C}\boldsymbol{\Gamma}^{\top})(\boldsymbol{B} \otimes \boldsymbol{\Gamma}\boldsymbol{C}\boldsymbol{\Gamma}^{\top} + v^2\boldsymbol{I}_{Nd_2})^{-1}\left(\text{vec}(\boldsymbol{Y}) - (\boldsymbol{I}_{d_2} \otimes \boldsymbol{\Gamma})\text{vec}(\boldsymbol{M})\right)$$
$$\boldsymbol{C}' = \boldsymbol{B} \otimes \boldsymbol{C} - (\boldsymbol{B} \otimes \boldsymbol{C}\boldsymbol{\Gamma}^{\top})(\boldsymbol{B} \otimes \boldsymbol{\Gamma}\boldsymbol{C}\boldsymbol{\Gamma}^{\top} + v^2\boldsymbol{I}_{Nd_2})^{-1}(\boldsymbol{B} \otimes \boldsymbol{\Gamma}\boldsymbol{C}),$$

*and*

$$\frac{d\nu'}{d\lambda}(d\boldsymbol{\omega}) = \mathcal{N}\left(\text{vec}(\boldsymbol{Y}_*) \mid (\boldsymbol{I}_{d_2} \otimes \boldsymbol{\Gamma}_*)\boldsymbol{m}'_{\boldsymbol{Y}}, v'^2\boldsymbol{I}_{N_*d_2} + (\boldsymbol{I}_{d_2} \otimes \boldsymbol{\Gamma}_*)\boldsymbol{C}'(\boldsymbol{I}_{d_2} \otimes \boldsymbol{\Gamma}_*^{\top})\right)$$
$$\mathcal{N}\left(\boldsymbol{\omega} \mid \boldsymbol{m}''_{\boldsymbol{Y},\boldsymbol{Y}_*}, \boldsymbol{C}''\right)\mathcal{N}_{\boldsymbol{Y}}, \quad \text{where}$$

$$\boldsymbol{m}''_{\boldsymbol{Y},\boldsymbol{Y}_*} = \boldsymbol{C}'(\boldsymbol{I}_{d_2} \otimes \boldsymbol{\Gamma}_*^\top)\big((\boldsymbol{I}_{d_2} \otimes \boldsymbol{\Gamma}_*)\boldsymbol{C}'(\boldsymbol{I}_{d_2} \otimes \boldsymbol{\Gamma}_*^\top) + v'^2\boldsymbol{I}_{N_*d_2}\big)^{-1}\big(\operatorname{vec}(\boldsymbol{Y}_*) - (\boldsymbol{I}_{d_2} \otimes \boldsymbol{\Gamma}_*)\boldsymbol{m}'_{\boldsymbol{Y}}\big)$$
$$+ \boldsymbol{m}'_{\boldsymbol{Y}}$$
$$\boldsymbol{C}'' = \boldsymbol{C}' - \boldsymbol{C}'(\boldsymbol{I}_{d_2} \otimes \boldsymbol{\Gamma}_*^\top)\big((\boldsymbol{I}_{d_2} \otimes \boldsymbol{\Gamma}_*)\boldsymbol{C}'(\boldsymbol{I}_{d_2} \otimes \boldsymbol{\Gamma}_*^\top) + v'^2\boldsymbol{I}_{N_*d_2}\big)^{-1}(\boldsymbol{I}_{d_2} \otimes \boldsymbol{\Gamma}_*)\boldsymbol{C}'.$$

*Here $\mathcal{N}_{\boldsymbol{Y}}$ hides an inconsequential closed-form factor which depends only on $\boldsymbol{Y}$. Hence the noisy posterior predictive distribution is available as $p(\boldsymbol{Y}_* \mid \boldsymbol{Y}, \boldsymbol{\Gamma}, \boldsymbol{\Gamma}_*) = \operatorname{Tr}(\boldsymbol{M}\boldsymbol{K}_{\nu',\psi}) / \operatorname{Tr}(\boldsymbol{M}\boldsymbol{K}_{\nu,\psi})$, whenever $\boldsymbol{K}_{\cdot,\psi}$ can be computed in closed-form for Gaussian base measures.*

**Noise-free posterior predictive**  In general, the noisy posterior predictive distribution $p(\boldsymbol{Y}_* \mid \boldsymbol{Y}, \boldsymbol{\Gamma}, \boldsymbol{\Gamma}_*)$ is not a GSF. However, if the posterior predictive distribution is evaluated on a large enough test set of size $N_* \geq d_1$, and the resulting feature matrix $\boldsymbol{\Gamma}_*$ is full-rank, then the noiseless posterior predictive distribution is itself a GSF.

**Corollary 8.** *Suppose additionally that $\boldsymbol{C}'$ and $\boldsymbol{\Gamma}_*$ are full-rank and $N_* \geq d_1$. Then*
$$\lim_{v'^2 \to 0+} \boldsymbol{m}''_{\boldsymbol{Y},\boldsymbol{Y}_*} = (\boldsymbol{I}_{d_2} \otimes \boldsymbol{\Gamma}_*)^\dagger \operatorname{vec}(\boldsymbol{Y}_*) = (\boldsymbol{I}_{d_2} \otimes (\boldsymbol{\Gamma}_*^\top\boldsymbol{\Gamma}_*)^{-1}\boldsymbol{\Gamma}_*^\top)\operatorname{vec}(\boldsymbol{Y}_*)$$
$$\lim_{v'^2 \to 0+} \boldsymbol{C}'' = \boldsymbol{0},$$

*and the noise free predictive distribution $P(d\boldsymbol{f}(\boldsymbol{X}_*) \mid \boldsymbol{Y}, \boldsymbol{\Gamma}, \boldsymbol{\Gamma}_*) = Q(d\boldsymbol{f}(\boldsymbol{X}_*) \mid \boldsymbol{M}, \nu'', \psi')$ belongs to a GSF with modified feature mapping and modified base measure,*

$$\psi'(\boldsymbol{f}(\boldsymbol{X}_*)) \triangleq \psi\Big((\boldsymbol{I}_{d_2} \otimes \boldsymbol{\Gamma}_*)^\dagger \boldsymbol{f}(\boldsymbol{X}_*)\Big) = \psi\Big(\operatorname{vec}\big(\boldsymbol{\Gamma}_*^\dagger \boldsymbol{f}(\boldsymbol{X}_*)\big)\Big)$$
$$\nu''(d\boldsymbol{f}(\boldsymbol{X}_*)) \triangleq \mathcal{N}\Big(\operatorname{vec}\big(\boldsymbol{f}(\boldsymbol{X}_*)\big) \mid (\boldsymbol{I}_{d_2} \otimes \boldsymbol{\Gamma}_*)\boldsymbol{m}'_{\boldsymbol{Y}}, (\boldsymbol{I}_{d_2} \otimes \boldsymbol{\Gamma}_*)\boldsymbol{C}'(\boldsymbol{I}_{d_2} \otimes \boldsymbol{\Gamma}_*^\top)\Big)d\boldsymbol{f}(\boldsymbol{X}_*). \quad (7)$$

*Proof.* Take the train likelihood to be multivariate Gaussian with homoskedastic noise with variance $v^2$,

$$p(\boldsymbol{Y} \mid \boldsymbol{\Gamma}\boldsymbol{\Omega}^\top) = \frac{1}{(2\pi v^2)^{d_1 d_2/2}} \exp\Big(-\frac{1}{2v^2}\operatorname{Tr}\big((\boldsymbol{Y} - \boldsymbol{\Gamma}\boldsymbol{\Omega}^\top)^\top(\boldsymbol{Y} - \boldsymbol{\Gamma}\boldsymbol{\Omega}^\top)\big)\Big)$$
$$= \frac{1}{(2\pi v^2)^{d_1 d_2/2}} \exp\Big(-\frac{1}{2v^2}\big(\operatorname{vec}(\boldsymbol{Y}) - (\boldsymbol{I}_{d_2} \otimes \boldsymbol{\Gamma})\boldsymbol{\omega}\big)^\top\big(\operatorname{vec}(\boldsymbol{Y}) - (\boldsymbol{I}_{d_2} \otimes \boldsymbol{\Gamma})\boldsymbol{\omega}\big)\Big)$$
$$= \mathcal{N}\big(\operatorname{vec}(\boldsymbol{Y}) \mid (\boldsymbol{I}_{d_2} \otimes \boldsymbol{\Gamma})\boldsymbol{\omega}, v^2\boldsymbol{I}_{Nd_2}\big)$$

Take the Radon-Nikodym derivative of the base measure $\mu$ with respect to Lebesgue measure $\lambda$ to be the density of a matrix Gaussian distribution,

$$\frac{d\mu}{d\lambda}(\boldsymbol{\Omega}) \triangleq p(\boldsymbol{\Omega}) = \frac{\exp\Big(-\frac{1}{2}\operatorname{Tr}\big(\boldsymbol{B}^\dagger(\boldsymbol{\Omega}^\top - \boldsymbol{M})^\top\boldsymbol{C}^\dagger(\boldsymbol{\Omega}^\top - \boldsymbol{M})\big)\Big)}{(2\pi)^{d_1 d_2/2}|\boldsymbol{B}|^{d_1/2}|\boldsymbol{C}|^{d_2/2}}$$
$$= \mathcal{N}(\boldsymbol{\omega} \mid \operatorname{vec}(\boldsymbol{M}), \boldsymbol{B} \otimes \boldsymbol{C})$$

We convert the product $p(\boldsymbol{Y} \mid \boldsymbol{\Gamma}\boldsymbol{\Omega}^\top)p(\boldsymbol{\Omega})$ into a product of two Gaussian probability density functions. In order to do so, we write $p(\boldsymbol{Y} \mid \boldsymbol{\Gamma}\boldsymbol{\Omega}^\top)p(\boldsymbol{\Omega}) = p(\boldsymbol{Y})p(\boldsymbol{\Omega} \mid \boldsymbol{Y}) = p(\boldsymbol{Y}, \boldsymbol{\Omega})$ and note that $\boldsymbol{Y}$ and $\boldsymbol{\Omega}$ are jointly Gaussian. Since $\boldsymbol{Y} = \boldsymbol{\Gamma}\boldsymbol{\Omega}^\top + \boldsymbol{E}$, the parameters of the marginal distribution of $\boldsymbol{Y}$ can be simply read off as

$$\mathbb{E}\big[\operatorname{vec}(\boldsymbol{Y})\big] = \operatorname{vec}(\boldsymbol{\Gamma}\boldsymbol{M}),$$
$$\mathbb{E}\big[\operatorname{vec}(\boldsymbol{Y})\operatorname{vec}(\boldsymbol{Y})^\top\big] = v^2\boldsymbol{I}_{Nd_2} + \mathbb{E}\big[\operatorname{vec}(\boldsymbol{\Gamma}\boldsymbol{\Omega}^\top)\operatorname{vec}(\boldsymbol{\Gamma}\boldsymbol{\Omega}^\top)^\top\big]$$
$$= v^2\boldsymbol{I}_{Nd_2} + \mathbb{E}\big[(\boldsymbol{I}_{d_2} \otimes \boldsymbol{\Gamma})\operatorname{vec}(\boldsymbol{\Omega}^\top)\operatorname{vec}(\boldsymbol{\Omega}^\top)^\top(\boldsymbol{I}_{d_2} \otimes \boldsymbol{\Gamma})^\top\big]$$
$$= v^2\boldsymbol{I}_{Nd_2} + (\boldsymbol{I}_{d_2} \otimes \boldsymbol{\Gamma})(\boldsymbol{B} \otimes \boldsymbol{C})(\boldsymbol{I}_{d_2} \otimes \boldsymbol{\Gamma}^\top)$$
$$= v^2\boldsymbol{I}_{Nd_2} + \boldsymbol{B} \otimes (\boldsymbol{\Gamma}\boldsymbol{C}\boldsymbol{\Gamma}^\top),$$

so that
$$p(\boldsymbol{Y}) = \mathcal{N}\big(\operatorname{vec}(\boldsymbol{Y}) \mid \underbrace{\operatorname{vec}(\boldsymbol{\Gamma}\boldsymbol{M})}_{(\boldsymbol{I}_{d_2} \otimes \boldsymbol{\Gamma})\operatorname{vec}(\boldsymbol{M})}, v^2\boldsymbol{I}_{Nd_2} + \boldsymbol{B} \otimes (\boldsymbol{\Gamma}\boldsymbol{C}\boldsymbol{\Gamma}^\top)\big).$$

For $p(\boldsymbol{\Omega} \mid \boldsymbol{Y})$, we compute the unnormalised posterior Gaussian probability density function by completing the square of the quadratic, following a typical calculation. This yields

$$p(\boldsymbol{\Omega} \mid \boldsymbol{Y}) = \mathcal{N}\big(\boldsymbol{\omega} \mid \boldsymbol{m}'_{\boldsymbol{Y}}, \boldsymbol{C}'_{\boldsymbol{Y}}\big),$$

where

$$\boldsymbol{m}'_{\boldsymbol{Y}} = \mathrm{vec}(\boldsymbol{M}) + (\boldsymbol{B} \otimes \boldsymbol{C}\boldsymbol{\Gamma}^\top)(\boldsymbol{B} \otimes \boldsymbol{\Gamma}\boldsymbol{C}\boldsymbol{\Gamma}^\top + v^2 \boldsymbol{I}_{Nd_2})^{-1}\big(\mathrm{vec}(\boldsymbol{Y}) - (\boldsymbol{I}_{d_2} \otimes \boldsymbol{\Gamma})\,\mathrm{vec}(\boldsymbol{M})\big)$$

$$\boldsymbol{C}'_{\boldsymbol{Y}} = \boldsymbol{B} \otimes \boldsymbol{C} - (\boldsymbol{B} \otimes \boldsymbol{C}\boldsymbol{\Gamma}^\top)(\boldsymbol{B} \otimes \boldsymbol{\Gamma}\boldsymbol{C}\boldsymbol{\Gamma}^\top + v^2 \boldsymbol{I}_{Nd_2})^{-1}(\boldsymbol{B} \otimes \boldsymbol{\Gamma}\boldsymbol{C}).$$

Therefore,

$$p(\boldsymbol{Y} \mid \boldsymbol{\Gamma}\boldsymbol{\Omega}^\top)\frac{d\mu}{d\lambda}(\boldsymbol{\Omega}) = \mathcal{N}\big(\mathrm{vec}(\boldsymbol{Y}) \mid (\boldsymbol{I}_{d_2} \otimes \boldsymbol{\Gamma})\,\mathrm{vec}(\boldsymbol{M}), v^2 \boldsymbol{I}_{Nd_2} + \boldsymbol{B} \otimes (\boldsymbol{\Gamma}\boldsymbol{C}\boldsymbol{\Gamma}^\top)\big)\mathcal{N}\big(\boldsymbol{\omega} \mid \boldsymbol{m}'_{\boldsymbol{Y}}, \boldsymbol{C}'_{\boldsymbol{Y}}\big),$$

(8)

Applying the same result again for $p(\boldsymbol{Y}_* \mid \boldsymbol{\Gamma}_*\boldsymbol{\Omega}^\top) = \mathcal{N}\big(\mathrm{vec}(\boldsymbol{Y}_*) \mid (\boldsymbol{I}_{d_2} \otimes \boldsymbol{\Gamma}_*)\boldsymbol{\omega}, v'^2 \boldsymbol{I}_{N_*d_2}\big)$, we may form the probability density function corresponding with the base measure $\nu'$,

$$p(\boldsymbol{Y}_* \mid \boldsymbol{\Gamma}_*\boldsymbol{\Omega}^\top)p(\boldsymbol{Y} \mid \boldsymbol{\Gamma}\boldsymbol{\Omega}^\top)\frac{d\mu}{d\lambda}(\boldsymbol{\Omega})$$
$$= \mathcal{N}\big(\mathrm{vec}(\boldsymbol{Y}) \mid (\boldsymbol{I}_{d_2} \otimes \boldsymbol{\Gamma})\,\mathrm{vec}(\boldsymbol{M}), v^2 \boldsymbol{I}_{Nd_2} + \boldsymbol{B} \otimes (\boldsymbol{\Gamma}\boldsymbol{C}\boldsymbol{\Gamma}^\top)\big)$$
$$\mathcal{N}\big(\mathrm{vec}(\boldsymbol{Y}_*) \mid (\boldsymbol{I}_{d_2} \otimes \boldsymbol{\Gamma}_*)\boldsymbol{m}'_{\boldsymbol{Y}}, v'^2 \boldsymbol{I}_{N_*d_2} + (\boldsymbol{I}_{d_2} \otimes \boldsymbol{\Gamma}_*)\boldsymbol{C}'_{\boldsymbol{Y}}(\boldsymbol{I}_{d_2} \otimes \boldsymbol{\Gamma}_*^\top)\big)\mathcal{N}\big(\boldsymbol{\omega} \mid \boldsymbol{m}''_{\boldsymbol{Y},\boldsymbol{Y}_*}, \boldsymbol{C}''_{\boldsymbol{Y},\boldsymbol{Y}_*}\big).$$

(9)

where

$$\boldsymbol{m}''_{\boldsymbol{Y},\boldsymbol{Y}_*} = \boldsymbol{m}'_{\boldsymbol{Y}} + \boldsymbol{C}'_{\boldsymbol{Y}}(\boldsymbol{I}_{d_2} \otimes \boldsymbol{\Gamma}_*^\top)\big((\boldsymbol{I}_{d_2} \otimes \boldsymbol{\Gamma}_*)\boldsymbol{C}'_{\boldsymbol{Y}}(\boldsymbol{I}_{d_2} \otimes \boldsymbol{\Gamma}_*^\top) + v'^2 \boldsymbol{I}_{N_*d_2}\big)^{-1}\big(\mathrm{vec}(\boldsymbol{Y}_*) - (\boldsymbol{I}_{d_2} \otimes \boldsymbol{\Gamma}_*)\boldsymbol{m}'_{\boldsymbol{Y}}\big)$$

$$\boldsymbol{C}''_{\boldsymbol{Y},\boldsymbol{Y}_*} = \boldsymbol{C}'_{\boldsymbol{Y}} - \boldsymbol{C}'_{\boldsymbol{Y}}(\boldsymbol{I}_{d_2} \otimes \boldsymbol{\Gamma}_*^\top)\big((\boldsymbol{I}_{d_2} \otimes \boldsymbol{\Gamma}_*)\boldsymbol{C}'_{\boldsymbol{Y}}(\boldsymbol{I}_{d_2} \otimes \boldsymbol{\Gamma}_*^\top) + v'^2 \boldsymbol{I}_{N_*d_2}\big)^{-1}(\boldsymbol{I}_{d_2} \otimes \boldsymbol{\Gamma}_*)\boldsymbol{C}'_{\boldsymbol{Y}}.$$

**Limiting base measure $\nu'$**  Let $\boldsymbol{C}'_{\boldsymbol{Y}} = \boldsymbol{L}\boldsymbol{L}^\top$ be a decomposition in terms of $\boldsymbol{L}$ such that $\boldsymbol{L} \in \mathbb{R}^{d_1 d_2 \times d_1 d_2}$ and suppose $\boldsymbol{C}'_{\boldsymbol{Y}}$ is invertible. We may use the limit definition of the pseudo-inverse to find

$$\lim_{v'^2 \to 0^+} \boldsymbol{C}'_{\boldsymbol{Y}}(\boldsymbol{I}_{d_2} \otimes \boldsymbol{\Gamma}_*^\top)\big((\boldsymbol{I}_{d_2} \otimes \boldsymbol{\Gamma}_*)\boldsymbol{C}'_{\boldsymbol{Y}}(\boldsymbol{I}_{d_2} \otimes \boldsymbol{\Gamma}_*^\top) + v'^2 \boldsymbol{I}_{N_*d_2}\big)^{-1}$$
$$= \lim_{v'^2 \to 0^+} \boldsymbol{L}\boldsymbol{L}^\top(\boldsymbol{I}_{d_2} \otimes \boldsymbol{\Gamma}_*^\top)\big((\boldsymbol{I}_{d_2} \otimes \boldsymbol{\Gamma}_*)\boldsymbol{L}\boldsymbol{L}^\top(\boldsymbol{I}_{d_2} \otimes \boldsymbol{\Gamma}_*^\top) + v'^2 \boldsymbol{I}_{N_*d_2}\big)^{-1}$$
$$= \boldsymbol{L}\big((\boldsymbol{I}_{d_2} \otimes \boldsymbol{\Gamma}_*)\boldsymbol{L}\big)^\dagger$$

The matrix $(\boldsymbol{I}_{d_2} \otimes \boldsymbol{\Gamma}_*) \in \mathbb{R}^{N_*d_2 \times d_1 d_2}$ has linearly independent columns, assuming $N_*$ is large enough. $\boldsymbol{L}$ has linearly independent rows (since $\boldsymbol{C}'_{\boldsymbol{Y}}$ is invertible). Therefore

$$\boldsymbol{L}\big((\boldsymbol{I}_{d_2} \otimes \boldsymbol{\Gamma}_*)\boldsymbol{L}\big)^\dagger$$
$$= \boldsymbol{L}\boldsymbol{L}^\dagger(\boldsymbol{I}_{d_2} \otimes \boldsymbol{\Gamma}_*)^\dagger$$
$$= (\boldsymbol{I}_{d_2} \otimes \boldsymbol{\Gamma}_*)^\dagger$$

We may then obtain the base measure $\lim_{v'^2 \to 0^+} \nu'(d\boldsymbol{\omega})$ which has parameters

$$\boldsymbol{l}_{\boldsymbol{Y},\boldsymbol{Y}^*} \triangleq \lim_{v'^2 \to 0^+} \boldsymbol{m}''_{\boldsymbol{Y},\boldsymbol{Y}_*} = (\boldsymbol{I}_{d_2} \otimes \boldsymbol{\Gamma}_*)^\dagger\,\mathrm{vec}(\boldsymbol{Y}_*)$$
$$\lim_{v'^2 \to 0^+} \boldsymbol{C}''_{\boldsymbol{Y},\boldsymbol{Y}_*} = \boldsymbol{0}.$$

We therefore have that

$$\lim_{v'^2 \to 0^+} \nu'(d\boldsymbol{\omega}) = \mathcal{N}\big(\mathrm{vec}(\boldsymbol{Y}) \mid (\boldsymbol{I}_{d_2} \otimes \boldsymbol{\Gamma})\,\mathrm{vec}(\boldsymbol{M}), v^2 \boldsymbol{I}_{Nd_2} + \boldsymbol{B} \otimes (\boldsymbol{\Gamma}\boldsymbol{C}\boldsymbol{\Gamma}^\top)\big)$$
$$\mathcal{N}\big(\mathrm{vec}(\boldsymbol{Y}_*) \mid (\boldsymbol{I}_{d_2} \otimes \boldsymbol{\Gamma}_*)\boldsymbol{m}'_{\boldsymbol{Y}}, (\boldsymbol{I}_{d_2} \otimes \boldsymbol{\Gamma}_*)\boldsymbol{C}'_{\boldsymbol{Y}}(\boldsymbol{I}_{d_2} \otimes \boldsymbol{\Gamma}_*^\top)\big)\boldsymbol{\delta}(\boldsymbol{\omega} - \boldsymbol{l}_{\boldsymbol{Y},\boldsymbol{Y}^*}).$$

Integration against this measure can be understood as an evaluation functional, and since $l_{Y,Y^*}$ is linear in $Y_*$, the evaluation is also GSF. More concretely,

$$z(\boldsymbol{\Theta}) \propto \int_{\Omega} \|\boldsymbol{\Theta}\psi(\boldsymbol{\omega})\|_2^2 \delta(\boldsymbol{\omega} - \boldsymbol{l}_{Y,Y^*})\, d\boldsymbol{\omega}$$
$$\mathcal{N}\big(\boldsymbol{f}(\boldsymbol{X}_*) \mid (\boldsymbol{I}_{d_2} \otimes \boldsymbol{\Gamma}_*)\boldsymbol{m}'_{\boldsymbol{Y}}, (\boldsymbol{I}_{d_2} \otimes \boldsymbol{\Gamma}_*)\boldsymbol{C}'_{\boldsymbol{Y}}(\boldsymbol{I}_{d_2} \otimes \boldsymbol{\Gamma}_*^{\top})\big)$$
$$= \|\boldsymbol{\Theta}\psi(\boldsymbol{l}_{Y,Y^*})\|_2^2 \mathcal{N}\big(\operatorname{vec}(\boldsymbol{f}(\boldsymbol{X}_*)) \mid (\boldsymbol{I}_{d_2} \otimes \boldsymbol{\Gamma}_*)\boldsymbol{m}'_{\boldsymbol{Y}}, (\boldsymbol{I}_{d_2} \otimes \boldsymbol{\Gamma}_*)\boldsymbol{C}'_{\boldsymbol{Y}}(\boldsymbol{I}_{d_2} \otimes \boldsymbol{\Gamma}_*^{\top})\big)$$
$$= \left\|\boldsymbol{\Theta}\psi\big((\boldsymbol{I}_{d_2} \otimes \boldsymbol{\Gamma}_*)^{\dagger}\operatorname{vec}(\boldsymbol{f}(\boldsymbol{X}_*))\big)\right\|_2^2$$
$$\mathcal{N}\big(\operatorname{vec}(\boldsymbol{f}(\boldsymbol{X}_*)) \mid (\boldsymbol{I}_{d_2} \otimes \boldsymbol{\Gamma}_*)\boldsymbol{m}'_{\boldsymbol{Y}}, (\boldsymbol{I}_{d_2} \otimes \boldsymbol{\Gamma}_*)\boldsymbol{C}'_{\boldsymbol{Y}}(\boldsymbol{I}_{d_2} \otimes \boldsymbol{\Gamma}_*^{\top})\big).$$

Hence the posterior predictive bleongs to a GSF with base measure $\nu''\big(d\boldsymbol{f}(\boldsymbol{X}_*)\big) = \mathcal{N}\Big(\operatorname{vec}\big(\boldsymbol{f}(\boldsymbol{X}_*)\big) \mid (\boldsymbol{I}_{d_2} \otimes \boldsymbol{\Gamma}_*)\boldsymbol{m}'_{\boldsymbol{Y}}, (\boldsymbol{I}_{d_2} \otimes \boldsymbol{\Gamma}_*)\boldsymbol{C}'_{\boldsymbol{Y}}(\boldsymbol{I}_{d_2} \otimes \boldsymbol{\Gamma}_*^{\top})\Big)d\boldsymbol{f}(\boldsymbol{X}_*).$ $\qquad\square$

# E  Algorithms

In response to reviewer feedback, we include a description of the algorithms used in our experiments. During the experiments, we run Algorithm 3 on the meta datset then run Algorithm 2 on the support/query dataset.

---

**Algorithm 1** Marginal likelihood (i.e., Proposition 2)

---

**Input:** Trainable final layer parameter $\boldsymbol{\Theta}$, trainable hidden network $\boldsymbol{\psi} : \mathbb{R}^d \to \mathbb{R}^n$, trainable base measure $\mu$, frozen likelihood $p(\boldsymbol{U} \mid \boldsymbol{\omega})$
**Output:** Marginal likelihood $p(\boldsymbol{U}) \in [0, \infty)$

1: Set $\boldsymbol{M} = \boldsymbol{\Theta}^\top \boldsymbol{\Theta}$.
2: Compute $\boldsymbol{K}_{\mu,\boldsymbol{\psi}}$        ▷ see Appendix B
3: Set $\nu(d\boldsymbol{\omega}) = p(\boldsymbol{U} \mid \boldsymbol{\omega})\mu(d\boldsymbol{\omega})$, then compute $\boldsymbol{K}_{\nu,\boldsymbol{\psi}}$        ▷ see Appendix B
4: Set $z(\boldsymbol{M}) = \mathrm{Tr}(\boldsymbol{M}\boldsymbol{K}_{\mu,\boldsymbol{\psi}})$        ▷ as per (1)
5: **return** $p(\boldsymbol{U}) = \mathrm{Tr}(\boldsymbol{M}\boldsymbol{K}_{\mu,\boldsymbol{\psi}})/z(\boldsymbol{M})$        ▷ as per Proposition 2

---

---

**Algorithm 2** GSFP posterior predictive update (i.e., Corollary 6)

---

**Input:** Frozen Gaussian likelihood $p(\boldsymbol{Y} \mid \boldsymbol{\gamma}(\boldsymbol{X})\boldsymbol{\omega})$ on support set. Query $\boldsymbol{X}^*$. Prior element of a GSF parameterised by: final layer parameter $\boldsymbol{\Theta} \in \mathbb{R}^{m \times n}$, hidden GSF network $\boldsymbol{\psi} : \mathbb{R}^d \to \mathbb{R}^n$, Gaussian base measure $\mu$, deep feature extractor $\boldsymbol{\gamma} : \mathbb{X} \to \mathbb{R}^d$.
**Output:** Posterior element of a GSF, parameterised by: final layer parameter, hidden network, and base measure.

1: Set $\boldsymbol{M} = \boldsymbol{\Theta}^\top \boldsymbol{\Theta}$.
2: Set $p(z \mid x^*)$ to be the Gaussian posterior predictive.        ▷ as per (6)
3: Set $\boldsymbol{\psi}'$        ▷ as per Corollary 6
4: Set $z(\boldsymbol{M}) = \mathrm{Tr}(\boldsymbol{M}\boldsymbol{K}_{\nu'',\boldsymbol{\psi}'})$        ▷ as per (1)
5: **return** Posterior element of a GSF parameterised by $\boldsymbol{M}, \nu''$ and $\boldsymbol{\psi}'$. That is, $P(d\boldsymbol{f} \mid \boldsymbol{M}, \nu'', \boldsymbol{\psi}') = \mathrm{Tr}(\boldsymbol{M}\boldsymbol{\psi}'(\boldsymbol{f})\boldsymbol{\psi}'(\boldsymbol{f})^\top)/z(\boldsymbol{M})\,\nu''(d\boldsymbol{f})$        ▷ as per (1) and Proposition 4

---

(Here $\boldsymbol{f}$ is shorthand for $\boldsymbol{f}(\boldsymbol{X}_*)$)

---

**Algorithm 3** GSFP Few-shot learning pretraining (i.e. paragraph 1 of Section 4)

---

**Input:** Meta-datasets $(\boldsymbol{X}_j, \boldsymbol{Y}_j)$ for $j = 1, \ldots, J$. Batch size $B \in \{1, \ldots, J\}$. Trainable final layer parameter $\boldsymbol{\Theta} \in \mathbb{R}^{m \times n}$. Trainable hidden network $\boldsymbol{\psi} : \mathbb{R}^d \to \mathbb{R}^n$. Trainable Gaussian base measure $\mu$. Deep feature extractor $\boldsymbol{\gamma} : \mathbb{X} \to \mathbb{R}^d$.
**Output:** Trained GSFP prior distribution

1: **for** each batch $b = 1, \ldots, \lfloor J/b \rfloor + 1$ **do**
2:      Set $\boldsymbol{U}_i = (\boldsymbol{Y}_i, \boldsymbol{\gamma}(\boldsymbol{X}_i))$, for each index $i$ in the batch. Compute Gaussian likelihood $p(\boldsymbol{Y}_i \mid \boldsymbol{\gamma}\boldsymbol{X}_i\boldsymbol{\omega}_i)$ as per (4), for each index $i$ in the batch
3:      Compute gradients of the marginal likelihood (Algorithm 1) on input $(\boldsymbol{\Theta}, \boldsymbol{\psi}, p(\boldsymbol{Y}_i \mid \boldsymbol{\gamma}(\boldsymbol{X}_i)\boldsymbol{\omega}))$ for each $i$ in the batch, with respect to parameters $\boldsymbol{\Theta}$ and parameters of $\boldsymbol{\psi}, \mu$ and $\boldsymbol{\gamma}$, using automatic differentation.
4:      Update $\boldsymbol{\Theta}$, and parameters of $\boldsymbol{\psi}, \mu$ and $\boldsymbol{\gamma}$, using the computed gradients (e.g. using SGD update rule).
5:      Optionally store gradient statistics (e.g. for Adam or other optimisers).
6: **end for**
7: **return** Trained GSFP prior distribution described by $\boldsymbol{\Theta}, \boldsymbol{\psi}, \mu$ and $\boldsymbol{\gamma}$.

---

# F    Experiment details - few-shot regression

Here we describe the six datasets (two of which come with licenses, the rest without), the implementation details, and the run times for each method.

## F.1    Sines

Sines dataset [Finn et al., 2017] is comprised of input $x$ from the interval $[-5, 5]$ with corresponding label $y = A \cdot \sin(x + p) + \epsilon$, where amplitude $A \sim U[0.1, 5.0]$ and phase $p \sim U[0, \pi]$ are uniformly distributed on their respective intervals, and $\epsilon \sim \mathcal{N}(0, 0.1)$ is a Gaussian noise with $0$ mean and a standard deviation of $0.1$.

In training, 10 input-label pairs are sampled with five each for support and query sets. Evaluation is performed on 500 inference iterations on 200 data points, with a 5/195 split ratio for support and query sets. Testing inputs are sampled from $[-5, 5]$ for in-distribution evaluation and $[-5, 10]$ for out-of-distribution evaluation.

## F.2    Mixed-Noise Sines

Mixed-Noise Sines [Sendera et al., 2021] is a variant of sines experiment by utilising input-dependent noise: $y = A \cdot \sin(x + p) + |x - p| \cdot \epsilon$ where $|\cdot|$ is an absolute value function.

## F.3    NDX100

NASDAQ100 Small Dataset [Qin et al., 2017] contains stock prices of 81 major corporations and index values of NASDAQ 100 from July 26 to December 22 in 2016. The data are recorded at a frequency of one point per minute, resulting in 390 data points collected in a standard trade day, with exceptions of 210 and 180 points on November 25 and December 22 separately.

Training sets and in-distribution evaluation sets are obtained by partitioning NASDAQ 100 index with a 70/30 split. Out-of-distribution evaluation is performed on the entire data series of Yahoo stock prices. During both training and evaluation, 10 input-label pairs are sampled with a fixed frequency of 1 point per 10 minutes. Both in-distribution and out-of-distribution evaluations report average performances over 500 sampled data sequences.

## F.4    EEG

EEG Steady-State Visual Evoked Potential Signals Dataset [Fernandez-Fraga et al., 2019] contains the electroencephalogram (EEG) data from 29 subjects performing different visual tests. The time-series data signals of various lengths (owing to different test durations) are recorded at a frequency of 128 Hz. This dataset is available under a Creative Commons Attribution 4.0 International (CC-BY-4.0) licence.

Electrode AF4 data obtained from the first Five Box Visual Test 1 on Subject 1 from Group A (file name A001SB1_1.csv) is used for training and in-distribution evaluation following a partition ratio of 70/30. Out-of-distribution evaluation is performed on the entire electrode AF4 data obtained from the first Five Box Visual Test 1 on Subject 3 from Group A (file name A003SB1_1.csv). During both training and evaluation, 10 input-label pairs are sampled with a frequency of 12.8 Hz. Both in-distribution and out-of-distribution evaluations report average performances over 500 sampled data sequences.

## F.5    QMUL

Queen Mary University of London Multiview Face Dataset (QMUL) [Gong et al., 1996] contains normalised facial images of 48 people. There are 133 facial images associated with each person, representing a slice of a view sphere where yaw and pitch span $\alpha \in \{10° \times i \, : \, i \in \{-9, -8, \ldots, 8, 9\}\}$ and $\beta \in \{10° \times j \, : \, j \in \{-3, -2, \ldots, 2, 3\}\}$ respectively.

We use the subset comprising of only 37 people, whose facial images are grayscale, for training and evaluation following a split ratio of 32/5. Randomly sampled head trajectory (a sequence of yaw-pitch pairs) is obtained as: $\{(\alpha, 10 \cdot \lfloor A \cdot \sin(\alpha/10 + p) + 3 \rfloor - 30)\}$, where amplitude $A \sim U[0.1, 5.0]$

and phase $\mathsf{p} \sim U[0, \pi]$ are uniformly distributed. Images and tilts along the trajectory are used as input-label pairs for training/evaluation. Note in out-of-distribution experiments, yaw is limited to a subset $\{10° \times i : i \in \{-9, -8, \ldots, -1, 0\}\}$ during training, but has the entire set in the evaluation. Both in-distribution and out-of-distribution evaluations report average performances over 500 sampled data sequences.

## F.6 Power

Power [Hebrail and Berard, 2006] dataset contains 2,075,259 electricity consumption readings in a house in Sceaux (France), spanning 47 months between December 2006 and November 2010. The time-series measurements of electricity consumption (watt per hour) are recorded at a frequency of one reading per minute on individual metres. This dataset is licenced under a Creative Commons Attribution 4.0 International (CC-BY-4.0) licence.

Readings from Sub_metering_3 are used for training and evaluation of the models. Following the setting in Sendera et al. [2021], 70% of the data, starting from the beginning of the time series, is used for both training and in-distribution evaluation, while the remaining 30% is used for out-of-distribution evaluation. During both training and evaluation, 10 input-label pairs are sampled with a frequency of 1 reading per 10 minutes. Both in-distribution and out-of-distribution evaluations report average performances over 500 sampled data sequences.

## F.7 Implementation details

GSFP is implemented using Pytorch and GPytorch frameworks. Features $\gamma(x)$ of inputs $x$ are obtained using a Multilayer Perceptron (MLP) in Sines, Mixed-Noise-Sines, NDX100 and EEG experiments, and a Convolutional Neural Network (CNN) for image data in QMUL experiment. The MLP has two fully-connected layers, with the first layer mapping $\mathbb{X} \to \mathbb{R}^d$ followed by a ReLU activation function [Hahnloser et al., 2000], and the second one mapping $\mathbb{R}^d \to \mathbb{R}^d$. The CNN has three convolutional layers with fixed kernel size of 3, stride of 2 and dilation of 2. Input channel number of the first convolutional layer is 3, and upsized to 36 in the two subsequent convolutional layers. Output channel number is 36 across all convolutional layers. ReLU activation function is appended to each of the first two convolutional layers. Output of the last convolutional layer is flattened into a $d$-dimensional vector. A scaled cosine function $\cos(\cdot)/\sqrt{d}$ is appended as the final activation.

The squared neural network consists of a hidden layer ($\psi : \Omega \to \mathbb{R}^n$, where $\Omega = \mathbb{R}^d$) and readout parameters ($\Theta \in \mathbb{R}^{m \times n}$). The hidden layer is a fully connected layer $\psi(\omega) = \sigma(W\omega + b)$ with a weight matrix $W \in \mathbb{R}^{n \times d}$ initialised from a standard normal distribution $\mathcal{N}(0, I)$, and a bias vector $b \in \mathbb{R}^n$ initialised to be all ones. For the activation function $\sigma$, we use the Snake activation function [Ziyin et al., 2020],

$$\text{Snake}_a(z) = z + \frac{1}{a}\sin^2(az) = z - \frac{1}{2a}\cos(2az) + \frac{1}{2az}.$$

The dimensions of data, latent feature and output used across the experiments are summarised in Table 3.

Table 3: Summary of dimensions used across the experiments. Recall that $d$ is the dimension of the feature $\gamma$ in the regression model $f(x) = \omega^\top \gamma(x)$ (see (2)), and $n$ and $m$ are the dimensions of the readout parameter $\Theta \in \mathbb{R}^{m \times n}$ in the GSF model (see § 2).

| Dimensions | Datasets | | | | | |
| --- | --- | --- | --- | --- | --- | --- |
| | Sines | Mixed-Noise Sines | NDX100 | EEG | Power | QMUL |
| $d$ | 40 | 40 | 5 | 5 | 5 | 2,916 |
| $n$ | 10 | 10 | 2 | 2 | 2 | 500 |
| $m$ | 10 | 10 | 1 | 1 | 1 | 500 |

Both DKT [Patacchiola et al., 2020] and NGGP [Sendera et al., 2021] are integrated into our code to facilitate their evaluations under the same experimental settings as those of GSFP. Their implementations follow the details specified in their original works, with only exceptions of increasing the dimension of the range of the feature extractor or feature mapping function ($\gamma$) from 1 to 5 in the NDX100, EEG, and Power datasets.

### F.8 Training details

All models, including DKT [Patacchiola et al., 2020], NGGP [Sendera et al., 2021] and GFSP, are trained with Adam optimiser [Kingma, 2014] using a fixed learning rate of $0.001$ and default beta coefficients of $\beta_1 = 0.9$ and $\beta_2 = 0.999$ across all experiments. All models are trained on $J = 10,000$ training datasets that are randomly sampled from the meta dataset as detailed in § F.1-F.6. The evaluation results of each meta model is averaged over 500 test datasets, with each entry representing the mean $\pm$ the standard deviation. Optimal performance of each meta model is obtained with grid search at a frequency of 500 training datasets. Both training and evaluation are conducted with a single RTX 2080TI GPU and an Intel(R) Xeon(R) Gold 6242 CPU @ 2.80GHz.

### F.9 Run times

Table 4: Benchmark results showing training time (mins) and inference speed (tests / s) on 1 meta dataset across regression tasks. The results are evaluated on 5 meta models, and each entry shows the mean $\pm$ standard deviation over these 5 meta models.

| Methods | | Training Time (mins) $\downarrow$ | | | | | |
|---|---|---|---|---|---|---|---|
| Name | Kernel | Sines | Mixed-Noise Sines | NDX100 | EEG | QMUL | Power |
| DKT | RBF | $1.04 \pm 0.02$ | $1.02 \pm 0.05$ | $1.05 \pm 0.02$ | $0.98 \pm 0.01$ | $3.06 \pm 0.11$ | $1.01 \pm 0.05$ |
| | Spectral | $1.07 \pm 0.03$ | $1.11 \pm 0.01$ | $1.16 \pm 0.01$ | $1.15 \pm 0.01$ | $3.34 \pm 0.01$ | $1.08 \pm 0.01$ |
| | NN Linear | $1.02 \pm 0.01$ | $0.98 \pm 0.03$ | $0.97 \pm 0.01$ | $0.95 \pm 0.01$ | $3.12 \pm 0.03$ | $0.98 \pm 0.01$ |
| NGGP | RBF | $41.59 \pm 0.60$ | $43.57 \pm 0.58$ | $53.41 \pm 2.59$ | $55.97 \pm 5.54$ | $57.68 \pm 0.88$ | $75.30 \pm 2.89$ |
| | Spectral | $37.40 \pm 1.41$ | $35.19 \pm 0.66$ | $53.57 \pm 0.87$ | $64.33 \pm 7.22$ | $57.13 \pm 1.41$ | $70.52 \pm 3.48$ |
| | NN Linear | $44.01 \pm 0.39$ | $42.53 \pm 1.65$ | $45.62 \pm 2.82$ | $63.75 \pm 5.21$ | $65.93 \pm 0.73$ | $61.92 \pm 3.57$ |
| GSFP | NN Linear | $4.09 \pm 0.32$ | $4.14 \pm 0.25$ | $4.39 \pm 0.04$ | $3.46 \pm 0.10$ | $9.67 \pm 0.02$ | $3.45 \pm 0.07$ |

| Methods | | Inference Speed (tests / s) $\uparrow$ | | | | | |
|---|---|---|---|---|---|---|---|
| Name | Kernel | Sines | Mixed-Noise Sines | NDX100 | EEG | QMUL | Power |
| DKT | RBF | $163.73 \pm 1.17$ | $167.84 \pm 1.98$ | $170.96 \pm 1.28$ | $174.98 \pm 1.88$ | $59.92 \pm 4.26$ | $169.38 \pm 0.44$ |
| | Spectral | $166.22 \pm 0.52$ | $163.67 \pm 0.64$ | $172.98 \pm 0.75$ | $175.61 \pm 1.44$ | $60.06 \pm 1.46$ | $174.97 \pm 3.25$ |
| | NN Linear | $184.35 \pm 1.46$ | $185.00 \pm 1.31$ | $195.49 \pm 0.66$ | $196.61 \pm 0.59$ | $62.25 \pm 1.97$ | $195.17 \pm 1.12$ |
| NGGP | RBF | $5.49 \pm 0.07$ | $5.30 \pm 0.05$ | $4.33 \pm 0.87$ | $4.11 \pm 0.46$ | $3.84 \pm 0.24$ | $3.85 \pm 0.28$ |
| | Spectral | $7.09 \pm 0.35$ | $6.80 \pm 0.85$ | $4.40 \pm 0.58$ | $4.33 \pm 0.13$ | $3.87 \pm 0.09$ | $3.78 \pm 0.16$ |
| | NN Linear | $4.96 \pm 0.27$ | $5.18 \pm 0.17$ | $5.86 \pm 0.94$ | $5.07 \pm 0.29$ | $3.06 \pm 0.16$ | $3.97 \pm 0.32$ |
| GSFP | NN Linear | $99.56 \pm 0.49$ | $99.08 \pm 0.97$ | $150.09 \pm 1.86$ | $149.61 \pm 1.31$ | $7.14 \pm 0.03$ | $149.41 \pm 1.80$ |

# G    Experimental details - vanilla regression

Here we describe the nine datasets selected by [Hernández-Lobato and Adams, 2015] for regression experiments, as well as the implementation details, experimental performances in terms of NLL, and run times for each method.

## G.1    Boston Housing

The Boston Housing dataset [Harrison Jr and Rubinfeld, 1978] consists of 506 instances with 14 variables describing the socioeconomic characteristics of neighbourhoods in Boston, Massachusetts. We use 13 features as input variables and the remaining variable, MEDV (median value of owner-occupied homes in $1000's), serves as the regression target. This dataset is licensed under an Apache 2.0 open source license.

## G.2    Concrete Compressive Strength

The Concrete Compressive Strength dataset [Yeh, 1998] consists of 1,030 instances with 8 variables describing the compositional structure, the curation duration, and the compressive strength of concrete samples. We use 7 features as input variables and the remaining variable, concrete compressive strength (MPa), serves as the regression target. This dataset is licensed under a Creative Commons Attribution 4.0 International (CC BY 4.0) license.

## G.3    Energy Efficiency

The Energy Efficiency dataset [Tsanas and Xifara, 2012] consists of 768 samples with 8 features describing the design details of the buildings. We use these 8 features as input variables and the variable, cooling load, serves as the regression target. This dataset is licensed under a Creative Commons Attribution 4.0 International (CC BY 4.0) license.

## G.4    Kin8nm

The Kin8nm dataset is part of the Kin family of datasets [Corke, 2002] that consists of 8,192 instances with 8 features simulating the forward kinetics of a robotic arm. We use these 8 features as input variables and the remaining variable (y) serves as the regression target.

## G.5    Condition Based Maintenance of Naval Propulsion Plants

The Condition Based Maintenance of Naval Propulsion Plant dataset [Coraddu et al., 2014] consists of 11,934 samples generated from a numerical simulation of a naval vessel with a gas turbine propulsion plant. We use the 16 steady-state sensor and operational measurements as input variables, and the compressor degradation serves as the regression target. This dataset is licensed under a Creative Commons Attribution 4.0 International (CC BY 4.0) license.

## G.6    Combined Cycle Power Plant

The Combined Cycle Power Plant dataset [Tüfekci, 2014] consists of 9,568 data points collected from a fully operational power plant over a period of 2006-2011. We use the hourly average readings from the four sensors as input variables and the remaining variable, net hourly electrical energy output (EP), serves as the regression target. This dataset is licensed under a Creative Commons Attribution 4.0 International (CC BY 4.0) license.

## G.7    Physicochemical Properties of Protein Tertiary Structure

The Physicochemical Properties of Protein Tertiary Structure dataset [Rana, 2013] consists of 45,730 samples of decoy structures. Each sample has nine physicochemical descriptor features, which we use as input variables. The remaining variable, root-mean-square deviation (RMSD), is the regression target. This dataset is licensed under a Creative Commons Attribution 4.0 International (CC BY 4.0) license. We use the first 15,000 samples for our regression experiment.

### G.8 Wine Quality

The Wine Quality dataset [Cortez et al., 2009] has two sub-datasets regarding red wine and white wine from the Portuguese "Vinho Verde" region. Following [Hernández-Lobato and Adams, 2015], we choose the red wine sub-dataset for our regression experiment. The red wine sub-dataset has 1,599 samples, each of which is described with 11 physicochemical features. The quality variable, taking ordinal integers ranging from 0 to 10, serves as the regression target. This dataset is licensed under a Creative Commons Attribution 4.0 International (CC BY 4.0) license.

### G.9 Yacht Hydrodynamics

The Yacht Hydrodynamics dataset [Gerritsma et al., 1981] consists of 308 samples with 6 features derived from hull geometry and hydrodynamic context. The output variable, resistance, serves as the regression target. This dataset is licensed under a Creative Commons Attribution 4.0 International (CC BY 4.0) license.

### G.10 Implementation details

Most of the implementation details for GSFP, DKT, and NGGP in the regression experiments follow those in Appendix F.7, with differences as follows. (a) The feature extractor is the two-layer MLP; (b) We fix $d = 5$, $n = 2$, and $m = 1$. Both (a) and (b) are applied across the nine regression experiments. We originally considered the same few-shot learning benchmark (all datasets) as in Sendera et al. [2021]. During the rebuttal period we applied DKT, NGGP and GSFP to vanilla regression datasets Boston, Concrete, Energy, Kin8nm, Naval, Power Plant, Protein, Wine and Yacht. In applying our multimodal, non-Gaussian model to vanilla regression, we repeat the message of Sendera et al. [2021]: "the main goal of [NNGP [Sendera et al., 2021]] was to show improvement of NGGP over standard GPs in the case of a few-shot regression task...Intuition is that NGGP may be superior to standard GPs in a simple regression setting for datasets with non-Gaussian characteristics, but do not expect any improvement otherwise". We stress that we did not perform hyperparameter tuning during the rebuttal period, and the results which follow are merely to demonstrate the performance of the models previously tuned for few-shot regression in vanilla regression settings. As such, we caution against general takeaways.

### G.11 Training details

All models, including DKT [Patacchiola et al., 2020], NGGP [Sendera et al., 2021] and GFSP, are trained with Adam optimiser [Kingma, 2014] using a fixed learning rate of $0.001$ and default beta coefficients of $\beta_1 = 0.9$ and $\beta_2 = 0.999$ across all experiments. The train/test split ratio of 80/20 is applied across the datasets as described in § G.1- G.9. Five models are trained for each method on each dataset using random seeds ranging from 1 to 5. All models are trained on the train set for 10,000 iterations. The evaluation result of each method is averaged over 5 models, with each entry representing the mean $\pm$ standard deviation. Optimal performance of each meta model is obtained with grid search at a frequency of 100 training iterations. Both training and evaluation are conducted with a single RTX 2080TI GPU and an Intel(R) Xeon(R) Gold 6242 CPU @ 2.80GHz.

### G.12 Results

As shown in Table 5, we observe that GSFP obtains the lower NLL on 7 out of 9 datasets, and the second lowest NLL on the Protein dataset. For Kin8nm, Naval, Power Plant and Protein datasets, DKT and NGGP with spectral kernels cannot be trained due to the large number of trainable parameters associated with the spectral kernel, which cannot fit in our available GPU memory.

### G.13 Run times

Table 5: Benchmark results showing test NLLs for regression tasks across 9 datasets described in § G.1-G.9. For each dataset, we train 5 models with different random seeds (1-5) via maximum marginal likelihood over the train set. At testing time, we condition on the train set and evaluate the NLL on the test set for each model. Each entry shows the mean ± standard deviation over 5 model evaluations. Best results in **bold**, second best underlined.

| Datasets | Methods | | | | | | |
| --- | --- | --- | --- | --- | --- | --- | --- |
| | DKT [Patacchiola et al., 2020] | | | NGGP [Sendera et al., 2021] | | | GSFP |
| | RBF | Spectral | NN Linear | RBF | Spectral | NN Linear | NN Linear |
| Boston | $0.335 \pm 0.198$ | $0.491 \pm 0.179$ | $0.478 \pm 0.129$ | $\underline{0.271 \pm 0.229}$ | $0.439 \pm 0.228$ | $0.288 \pm 0.084$ | $\mathbf{-0.345 \pm 0.272}$ |
| Concrete | $\underline{0.300 \pm 0.123}$ | $0.302 \pm 0.135$ | $0.442 \pm 0.082$ | $0.647 \pm 0.255$ | $0.447 \pm 0.247$ | $0.411 \pm 0.407$ | $\mathbf{-0.072 \pm 0.175}$ |
| Energy | $-0.302 \pm 0.173$ | $-0.277 \pm 0.094$ | $\underline{-0.474 \pm 0.118}$ | $0.481 \pm 0.135$ | $0.597 \pm 0.027$ | $0.364 \pm 0.107$ | $\mathbf{-1.342 \pm 0.393}$ |
| Kin8nm | $\underline{0.153 \pm 0.011}$ | - | $0.388 \pm 0.033$ | $0.310 \pm 0.150$ | - | $0.655 \pm 0.113$ | $\mathbf{0.021 \pm 0.080}$ |
| Naval | $\underline{-1.070 \pm 0.011}$ | - | $0.994 \pm 0.156$ | $0.405 \pm 0.474$ | - | $0.830 \pm 0.135$ | $\mathbf{-2.019 \pm 0.219}$ |
| Power Plant | $\underline{-0.220 \pm 0.193}$ | - | $-0.028 \pm 0.022$ | $-0.114 \pm 0.137$ | - | $-0.156 \pm 0.136$ | $\mathbf{-0.552 \pm 0.137}$ |
| Protein | $1.024 \pm 0.045$ | - | $1.144 \pm 0.024$ | $\mathbf{0.418 \pm 0.048}$ | - | $0.844 \pm 0.097$ | $\underline{0.818 \pm 0.152}$ |
| Wine | $1.162 \pm 0.034$ | $1.192 \pm 0.050$ | $1.170 \pm 0.031$ | $-1.381 \pm 0.264$ | $\underline{-1.444 \pm 0.276}$ | $\mathbf{-1.666 \pm 0.664}$ | $0.755 \pm 0.138$ |
| Yacht | $-1.850 \pm 0.251$ | $-1.278 \pm 1.219$ | $-1.496 \pm 0.256$ | $\underline{-2.350 \pm 0.534}$ | $0.542 \pm 0.159$ | $0.166 \pm 0.591$ | $\mathbf{-2.873 \pm 0.238}$ |

Table 6: Benchmark results showing training time (hours) and inference speed (tests / s) for each model across regression tasks. The results are evaluated on 5 models, and each entry shows the mean ± standard deviation over these 5 models.

| Training Time (hours) ↓ | | | | | | | |
| --- | --- | --- | --- | --- | --- | --- | --- |
| Datasets | Methods | | | | | | |
| | DKT [Patacchiola et al., 2020] | | | NGGP [Sendera et al., 2021] | | | GSFP |
| | RBF | Spectral | NN Linear | RBF | Spectral | NN Linear | NN Linear |
| Boston | $0.034 \pm 0.001$ | $0.037 \pm 0.001$ | $0.032 \pm 0.002$ | $5.042 \pm 0.619$ | $4.024 \pm 0.198$ | $4.828 \pm 0.353$ | $0.100 \pm 0.003$ |
| Concrete | $0.091 \pm 0.007$ | $0.112 \pm 0.005$ | $0.115 \pm 0.011$ | $10.660 \pm 1.209$ | $10.592 \pm 1.400$ | $9.229 \pm 1.258$ | $0.176 \pm 0.003$ |
| Energy | $0.048 \pm 0.000$ | $0.053 \pm 0.001$ | $0.045 \pm 0.001$ | $7.585 \pm 0.705$ | $7.741 \pm 0.645$ | $6.866 \pm 0.450$ | $0.121 \pm 0.004$ |
| Kin8nm | $1.923 \pm 0.016$ | - | $0.727 \pm 0.026$ | $62.107 \pm 7.461$ | - | $64.435 \pm 5.747$ | $1.238 \pm 0.002$ |
| Naval | $3.149 \pm 0.075$ | - | $1.367 \pm 0.052$ | $127.623 \pm 11.349$ | - | $108.597 \pm 5.025$ | $2.025 \pm 0.008$ |
| Power Plant | $1.664 \pm 0.271$ | - | $0.839 \pm 0.061$ | $76.962 \pm 4.722$ | - | $68.406 \pm 5.127$ | $1.519 \pm 0.003$ |
| Protein | $5.415 \pm 0.559$ | - | $2.303 \pm 0.092$ | $144.444 \pm 10.479$ | - | $111.314 \pm 12.836$ | $2.868 \pm 0.020$ |
| Wine | $0.099 \pm 0.008$ | $0.175 \pm 0.017$ | $0.101 \pm 0.008$ | $16.680 \pm 2.621$ | $18.150 \pm 3.623$ | $18.791 \pm 4.215$ | $0.206 \pm 0.001$ |
| Yacht | $0.033 \pm 0.000$ | $0.035 \pm 0.000$ | $0.032 \pm 0.000$ | $4.677 \pm 0.108$ | $4.432 \pm 0.206$ | $4.867 \pm 0.211$ | $0.109 \pm 0.001$ |

| Inference Time (seconds) ↓ | | | | | | | |
| --- | --- | --- | --- | --- | --- | --- | --- |
| Datasets | Methods | | | | | | |
| | DKT [Patacchiola et al., 2020] | | | NGGP [Sendera et al., 2021] | | | GSFP |
| | RBF | Spectral | NN Linear | RBF | Spectral | NN Linear | NN Linear |
| Boston | $0.30 \pm 0.00$ | $0.30 \pm 0.00$ | $0.27 \pm 0.00$ | $7.09 \pm 0.53$ | $7.36 \pm 0.07$ | $6.85 \pm 1.22$ | $0.52 \pm 0.01$ |
| Concrete | $1.10 \pm 0.00$ | $1.43 \pm 0.05$ | $1.55 \pm 0.17$ | $14.11 \pm 2.33$ | $12.95 \pm 1.55$ | $15.74 \pm 1.49$ | $1.04 \pm 0.00$ |
| Energy | $0.70 \pm 0.00$ | $0.79 \pm 0.04$ | $0.55 \pm 0.00$ | $8.20 \pm 0.82$ | $8.38 \pm 0.93$ | $8.64 \pm 0.03$ | $1.02 \pm 0.02$ |
| Kin8nm | $30.96 \pm 0.19$ | - | $20.54 \pm 1.34$ | $118.48 \pm 11.49$ | - | $113.30 \pm 12.79$ | $8.53 \pm 0.01$ |
| Naval | $67.41 \pm 2.61$ | - | $42.08 \pm 3.60$ | $178.25 \pm 3.59$ | - | $155.69 \pm 2.97$ | $15.03 \pm 0.19$ |
| Power Plant | $42.51 \pm 0.56$ | - | $27.32 \pm 2.16$ | $144.27 \pm 4.71$ | - | $134.66 \pm 10.57$ | $9.67 \pm 0.02$ |
| Protein | $118.11 \pm 0.02$ | - | $70.83 \pm 2.72$ | $297.26 \pm 14.70$ | - | $211.55 \pm 4.89$ | $15.10 \pm 0.03$ |
| Wine | $1.70 \pm 0.00$ | $2.98 \pm 0.10$ | $1.62 \pm 0.00$ | $17.08 \pm 0.72$ | $25.11 \pm 1.95$ | $21.76 \pm 5.18$ | $2.10 \pm 0.01$ |
| Yacht | $0.17 \pm 0.00$ | $0.17 \pm 0.00$ | $0.15 \pm 0.00$ | $5.59 \pm 1.32$ | $6.45 \pm 0.74$ | $5.24 \pm 1.66$ | $0.41 \pm 0.01$ |

