# OpenReview forum: "Squared families are useful conjugate priors"
_NeurIPS.cc/2025/Conference — NeurIPS 2025 poster_

### Official Review · Reviewer_7C3F · 2025-06-27

**Clarity:** 1
**Significance:** 2
**Originality:** 3
**Rating:** 4
**Confidence:** 2

**Summary:**

The paper discusses the use of squared families as priors in Bayesian inference.
This work considers the generalised square family (GSF) of distributions,
showing that it is an especially convenient family for Bayesian priors, because
(under some constraints) (1) the marginal likelihood and posterior predictive
become tractable, (2) the posterior is also GSF. These properties make GSF a good choice
for Bayesian regression, and in the experiment section, the authors demonstrate
that GSF priors can be used to outperform existing methods (DKT and NGGP) on
few-shot learning benchmarks.

**Questions:**

- You say that GSF is a rich family of distributions, but then (correct me if
  I'm wrong) experiments are only conducted using Gaussian
  measures/likelihoods. Are you able to try with any alternatives? Specifically,
  the categorical likelihood would be very useful.
- Are you able to combine GSF with e.g. deeper feature extractors? 1 layer is a
  bit artificial. Results would be more convincing if they included a more
  thorough hyperparameter search.
- How exactly (algorithmically) are you performing your Bayesian updates and NN
  updates? An algorithmic description in the experimental section would help
  ellucidate this.

**Ethical Concerns:**

["NO or VERY MINOR ethics concerns only"]

**Final Justification:**

During rebuttals, authors addressed my concerns, providing new experiments, comprehensive examples of when GSFs are tractable, improved related work (in another review), and a plan for improving the introduction/background. Due to this, I will increase my score.

**Limitations:**

Limitations are adequately discussed

**Paper Formatting Concerns:**

None of the pdf links function correctly!

**Quality:**

3

**Strengths And Weaknesses:**

Strenghts:

- The paper has strong results, and comprehensively demonstrates the benefits of
  GSF priors in Section 3.
- The experiments cleanly show benefits of GSF over alternative few-shot
  learning methods in Table 1 and Figure 4.

Weaknesses:

- The introduction, especially the first subsection of squared families, reads
  more like a background section, lacking a clear problem statement and
  motivation for the use of GSF priors.
- The contributions are not clear in the introduction. Two examples: first, the few-shot setting used
  in the experiments is not disucssed at all; second, authors say 'we first
  construct a hierarchy of families, ... (GSF)', which on first reading sounds
  like a contribution, but then confusingly GSFs appear in the background section.
- Claims that GSF is a rich and 'convenient to work with' family of
  distributions (e.g. line 100/101, line 164) is not easily verifable from the
  main text. Concrete examples would be helpful for readers.
- The experimental results could be more comprehensive, e.g. by using
  additional, more modern datasets/methods for comparison
- Experiments are conducted only with a single layer of MLP. I would imagine DKT
  and NGGP could be improved with deeper networks, potentially leading the reader to doubt the baselines.
- Figure 4 could be much clearer, e.g. by labelling rows/columns and axes and/or
  adding a legend.
- There are no functioning links in the paper (e.g. figure links, table links, refs, etc.)

---

> ### Author Rebuttal · Authors · 2025-07-25
>
> Thanks for the detailed review. Please let us know if you have additional queries.
>
> ## 1. Intro reads like a background. Few-shot learning is not discussed
>
> We will add further details to the few-shot learning setting to the end of the introduction. We will move most of the first paragraphs of section 4 (adapting the text where required) to the end of section 1 in a new few-shot learning subsection. We will add motivation in terms of difficulty of Bayesian inference (see response 1. to Reviewer Z38J). Please read this response - we clarify that our main contributions concern Bayesian inference generally. Our problem statement is then to do fast and expressive closed-form Bayesian inference, and we use the tool of GSFs. As an application, we consider few-shot learning.
>
> ---
>
> ## 2. Definition 1 (GSFs) in background
>
> We agree with your suggestion that this belongs in the same section as our contributions. We will move definition 1 to the start of section 3, in a new subsection. This just requires moving the heading of section 3 back a couple of paragraphs.
>
> ---
>
> ## 3. Concrete examples (rich and convenient to work with)
>
> Using extra space afforded by Reviewer sKy9's suggestions as well as the extra page, we will add a short table in the main paper as well as an extended paper in the appendix which shows concrete examples of squared families with closed form normalising constant (referring to it in line 100/101 and 164).
>
> | Domain $\Omega$ |   Base measure $\mu$ | Feature $\psi(\omega) $| Reference and/or Notes |
> | ---- | ---- | ---- | ---- |
> |  |  | NN hidden layer | As per Appendix B.3. **For example:** |
> | | | | |
> | $\mathbb{R}^d$  | $\mathcal{N}(\text{mean},\text{Cov})$ | $\text{erf}(\mathbf{W} \omega)$ | Williams [1996] |
> | $\mathbb{R}^d$  | $\mathcal{N}(\text{mean},\text{Cov})$ | $\text{ReLU}(\mathbf{W} \omega)$ | Cho and Saul [2009] |
> | $\mathbb{R}^d$  | $\mathcal{N}(\text{mean},\text{Cov})$ | $\text{cos}(\mathbf{W} \omega + \mathbf{b})$ | Pearce et al. [2020] |
> | $\mathbb{R}^d$  | $\mathcal{N}(\text{mean},\text{Cov})$ | $\text{GELU}(\mathbf{W} \omega)$ | Tsuchida et al. [2021] |
> | $\mathbb{R}^d$  | $\mathcal{N}(\text{mean},\text{Cov})$ | $\text{Snake}(\mathbf{W} \omega)$ | Tsuchida et al. [2023] |
> | | | | |
> | | | Exponential | As per Appendix B.1. **For example:** |
> | | | | |
> | $\mathbb{R}^d$  | $\mathcal{N}(\mathbf{0},\mathbf{I})$    | $\exp(\mathbf{W} \omega+\mathbf{b})$ | e.g. Nielsen and Garcia [2024] |
> | $\{ 0, 1, 2,\ldots \}$  | $(\omega !)^{-1} \nu$    | $\exp(\mathbf{W} \omega+\mathbf{b})$ | e.g.  Nielsen and Garcia [2024] |
> | $\mathbb{S}^{d-1}$  | Uniform    | $\exp(\mathbf{W} \omega+\mathbf{b})$ | e.g.  Brown [1986] |
>
> (**Caption:** All of these settings admit closed-form squared family kernels $\matrix{K}_{\mu,\psi}$ in linear time in the dimension $d$. Here erf denotes the error function, $\mathbb{S}^{d-1}$ denotes the unit hypersphere, Snake denotes the snake activation function and $\nu$ denotes the counting measure. Further examples are given in Han et al. (Table 1, 2022) and Tsuchida et al. (Table 1, 2023).)
>
> (Note: $\Omega$ is the domain of the distribution. Openreview does not allow typesetting as in paper.)
>
> We are also happy to type some of the equations of the closed-form kernels in the text. Please let us know if you would like this as well. To give you an idea of what this would look like, for the Cho and Saul [2009] row in the table above, the $ij$th entry $ \mathbf{K}_{\mu,\psi}(i,j)$ is
> $$ \mathbf{K}_{\mu,\psi}(i,j) = \frac{\Vert \mathbf{w}_i \Vert \Vert \mathbf{w}_j \Vert }{2\pi}\left( \sin  \rho - (\pi - \rho) \cos\rho  \right), $$
> where $\rho$ is the angle between $\mathbf{w}_i $ and $\mathbf{w}_j $, i.e. $\rho = \cos^{-1} \frac{\mathbf{w}_i^\top \mathbf{w}_j  }{\Vert \mathbf{w}_i \Vert \Vert \mathbf{w}_j \Vert}$. The complexity of this calculation is only linear in the dimension $d$ of the distribution (which is the same as the number of entries in $\mathbf{w}_i$ or $\mathbf{w}_j$), because we just need to compute inner products and norms. The same linear complexity holds for all examples in the table.
>
> ---
>
> ## 4. Other methods
>
> We compared with probabilistic methods for function inference (GPs/DKTs and NGGPs) with exact normalising constant. These are the only competitor methods we are aware of with these properties. Sendera et al. only compared DKT/NGGP. If you can suggest other methods with these properties, we may be able to add if we have time, or at the very least discuss similarities/differences/strengths/weaknesses.
>
> ---
>
> ## 5. Other datasets
>
> We used all of the same datasets as Sendera et al. for few shot learning. See our response to Reviewer sKy9 for **9 additional** experiments on vanilla regression, with standard UCI datasets.
>
> ---
>
> ## 6. Number of layers
>
> Note that experiments are **not** conducted with only a single layer MLP. There is a distinction between the feature mapping $\mathbf{\gamma}$ (which is a deep network, common to all methods, and may be a 3 hidden layer conv net or 2 hidden layer MLP depending on the dataset) and the function $\psi$ which only appears in the GSF, and is a single hidden layer network. See Appendix E.7. Note that we use the same feature mappings for DKT and NGPP as in respective original papers (Patacchiola et al., 2020 and Sendera et al., 2021].
>
> ---
>
> ## 7. Links
>
> Apologies, for this oversight. The final paper will have functioning links.
>
> ---
>
> ## 8. GSF are rich claim
>
> Our claim is that the GSF priors/posteriors are rich, i.e. they can model non-Gaussian and multimodal distributions. This is not a statement about likelihoods - indeed, we do not claim that GSFs are conjugate to all likelihoods. In the case of regression, we rely on a Gaussian likelihood and our results follow from its conjugacy with the Gaussian base measure -- but note that the priors and posteriors have a complex neural-network based non-Gaussian density with respect to that base measure (this is what makes them expressive while preserving conjugacy). We also analyse other likelihoods (e.g. Poisson in Figure 1, others in Appendix B).
>
> ---
>
> ## 9. Algorithms
>
> We will add the following algorithms to the paper (Formatted better in Latex Openreview doesn't allow some things. We skip some  algorithms here due to space, but these are all that are required for GSFP few shot learning). In section 4, we run algorithm 3 on the meta dataset then algorithm 2 on the support/query dataset.
>
> **Alg. 1:** ``Marginal likelihood`` (i.e. Proposition 2). **Inputs:** Trainable final layer parameter $\mathbf{\Theta} \in \mathbb{R}^{m \times n}$, Trainable hidden network $\psi:\mathbb{R}^d \to \mathbb{R}^n$, Trainable base measure $\mu$, Frozen likelihood $p(\mathbf{U} \mid \omega)$. **Output:** Marginal likelihood $p(\mathbf{U}) \in [0,\infty)$.
>
> 0. Set $\mathbf{M} = \mathbf{\Theta}^\top\mathbf{\Theta}$.
> 1. Compute $\mathbf{K}_{\mu,\psi}$ (see Appendix B, enlarged with response to Question 4 above).
> 2. Set $\nu(d\omega) = p(\mathbf{U} \mid \omega) \mu(d \omega)$. Compute $\mathbf{K}_{\nu,\psi}$ (see Appendix B, enlarged with response to Question 4 above).
> 3. Set $z(\mathbf{M}) = \text{Tr}(\mathbf{M} \mathbf{K}_{\mu,\psi})$, as per (1).
> 4. Return $p(\mathbf{U})=\text{Tr}(\mathbf{M} \mathbf{K}_{\nu,\psi})/z(\mathbf{M}),$ as per Proposition 2.
>
> **Alg. 2:** ``GSFP posterior predictive update`` (i.e. Corollary 7). **Inputs:** Frozen Gaussian likelihood $p(\mathbf{Y} \mid  \gamma(\mathbf{X}) \omega)$ on support set. Query $\mathbf{X}_\ast$. Prior element of a GSF, parameterised by: Final layer parameter $\mathbf{\Theta} \in \mathbb{R}^{m \times n}$, Hidden GSF network $\mathbf{\psi}:\mathbb{R}^d \to \mathbb{R}^n$, Gaussian Base measure $\mu$, Deep feature extractor $\gamma:\mathbb{X} \to \mathbb{R}^d$ **Output:** Posterior element of a GSF, parameterised by: Final layer parameter, Hidden network and Base measure.
>
> 0. Set $\mathbf{M} = \mathbf{\Theta}^\top\mathbf{\Theta}$.
> 1. Set $\nu''(d\omega)$ to be Gaussian posterior predictive, as per (6).
> 2. Set $\mathbf{\psi}'$ as per Corollary 7.
> 2. Set $z(\mathbf{M}) = \text{Tr}(\mathbf{M} \mathbf{K}_{\nu'',\mathbf{\psi}'})$, as per (1).
> 3. Return Posterior element of a GSF, parameterised by $\mathbf{M},\nu''$ and $\mathbf{\psi}'$. That is,
> $P(d \mathbf{f} \mid \mathbf{M},\nu'',\mathbf{\psi}') =\Big( \text{Tr}( \mathbf{M} \psi'( {\mathbf{f}} ) \psi'( {\mathbf{f}} )^\top )/z(\mathbf{M}) \Big)\nu''(d\mathbf{f})$, as per (1) and Proposition 2.
>
> ($\mathbf{f}$ is shorthand for $\mathbf{f}(\mathbf{X}_\ast)$).
>
> **Alg. 3:** ``GSFP Few-shot learning pretraining`` (i.e. paragraph 1 of section 4). **Inputs:** Meta datasets $(\mathbf{X}_j,\mathbf{Y}_j)$ for $j = 1,\ldots, J$, Batch size $B \in \{1,\ldots,J\}$, Trainable final layer parameter $\mathbf{\Theta} \in \mathbb{R}^{m \times n}$, Trainable hidden network $\mathbf{\psi}:\mathbb{R}^d \to \mathbb{R}^n$, Trainable Gaussian base measure $\mu$, Deep feature extractor $\gamma:\mathbb{X} \to \mathbb{R}^d$ **Output:** Trained GSFP prior distribution.
>
> 0. For batch $b = 1,\ldots,\lfloor J/B \rfloor + 1$:
>
>     0.a Set $\mathbf{U}_i = (\mathbf{Y}_i,\gamma(\mathbf{X}_i) )$, for each index $i$ in the batch. Compute Gaussian likelihood $p(\mathbf{Y}_i \mid  \gamma(\mathbf{X}_i) \omega)$, as per (4), for each index $i$ in the batch.
>
>     0.b Compute gradients of ``Marginal likelihood``($\mathbf{\Theta},\mathbf{\psi},\mu, p(\mathbf{Y}_i \mid  \gamma(\mathbf{X}_i) \omega)$) for each $i$ in the batch with respect to $\mathbf{\Theta}$ and parameters of $\mathbf{\psi},\mu$ and $\gamma$ using automatic differentiation.
>
>     0.c Update $\mathbf{\Theta}$ and parameters of $\mathbf{\psi},\mu$ and $\gamma$ using gradients (e.g. using SGD update rule).
>
>     0.d Store gradient information if required (e.g. for Adam etc.).
>
> 1. Return $\mathbf{\Theta},\mathbf{\psi},\mu,\gamma$.
>
> ---
>
> ## References
>
> Brown (1986). Fundamentals of statistical exponential families: with applications in statistical decision theory.

---

> ### Comment · Reviewer_7C3F · 2025-08-05
>
> Thank you for taking the time to address feedback in my review.
>
> I believe the addition of new datasets and detailed algorithms is a good step forward. However, I'm still worried that the introduction/background is not going to be clear enough, please see below.
>
> >Our problem statement is then to do fast and expressive closed-form Bayesian inference, and we use the tool of GSFs.
>
> It's not clear if you will be making sure GSFs are well motivated.
> Please make sure to place GSFs in context of Bayesian inference as a whole,
> why/when
> we should use them vs. other methods etc.. The problem statement you described
> sounds very broad -- it is very general to 'do fast and expressive
> closed-form Bayesian inference'. Your problem statement could be catered more to
> the few-shot learning, and then the application is an even more specific slice.
>
> ---
>
> > Concrete examples (rich and convenient to work with)
>
> I really appreciate the concrete examples for when normalizing constants can be
> computed in closed form.
>
> >See our response to Reviewer sKy9 for 9 additional experiments on vanilla regression, with standard UCI datasets.
>
> Thank you for compiling some further benchmarks on extra datasets. The results
> look fairly convincing, though I worry some of the UCI datasets are quite small.
>
> > and may be a 3 hidden layer conv net or 2 hidden layer MLP depending on the dataset)
>
> Thank you for clarifying that it is not just 1 layer MLP. These are still very
> small networks however.

---

> > ### Author Response · Authors · 2025-08-07
> >
> > Thank you for the further suggestions on improving clarity.
> >
> > > Please make sure to place GSFs in context of Bayesian inference as a whole, why/when we should use them vs. other methods etc.
> >
> > We will make sure to place GSFs in the context of Bayesian inference as a whole, using the text in our response to Reviewer Z38J's first comment. This also includes a discussion on when to use them versus other methods. To further add to this, to the best of our knowledge, only exponential families and what Christian Robert (2007) calls "quasi-exponential families" allow for closed-form inference in conjugate families. Furthermore, such models are not as rich as neural network based models, and cannot represent a wide range of posterior distributions.
> >
> > Based on your feedback, the problem statement will be formulated as follows:
> > 1. We first solve the problem of fast and expressive closed-form Bayesian inference using the tool of GSFs.
> > 2. (Noting the desiderata of few-shot learning, which we motivate earlier as per our response 1 to your initial suggestions, i.e.
> > "We will move most of the first paragraphs of section 4 (adapting the text where required) to the end of section 1 in a new few-shot learning subsection"), We investigate whether the solution to the problem above can be used for few-shot learning.
> >
> > Please let us know if this meets your expectation.

---

> > > ### Comment · Reviewer_7C3F · 2025-08-07
> > >
> > > Thank you for these clarifications. Overall the rebuttal the authors' rebuttal was fairly comprehensive, not least due to the new experiments and the related work shown in reviewer 2rpL's rebuttal. Under the proviso that the introduction/background will written more clearly I will increase my score

---

### Official Review · Reviewer_sKy9 · 2025-06-28

**Clarity:** 2
**Significance:** 2
**Originality:** 3
**Rating:** 3
**Confidence:** 3

**Summary:**

This paper explores *Squared Families* as prior distributions and the conjugacy of prior distributions belonging to *Generalized Squared Families*. GSFs are indexed by a parameter, $\textbf{M}$, a base measure, $\mu$, and a feature, $\Phi$. The authors show GSF priors can admit closed form expressions for commonly used objects (e.g. marginal likelihoods, posterior predictive distributions) when, for example, the base measure is conjugate to the likelihood. The paper explores particular settings (base measure + likelihood pairs) and works through results for those settings. The paper concludes with an experiments section comparing against two baselines on a collection of datasets.

**Questions:**

1. The computational benefits of GSPs were presented as one of their sources of value, but section 5's limitations makes me question if this is actually true since the complexity presented on lines 269-271 seem comparable to naive GPs. Can the authors please explain what in particular makes the reported complexities desirable? These seem as restrictive as any kernel based method.

2. The paper presents a substantial amount of material, but it is not as accessible as it could be for the work to have broader impact. For instance, Section 3 could be shortened, as the key point about conjugacy for exponential families is already made immediately after Proposition 4. The content in Sections 3.2 and 3.3 could be streamlined, with technical details moved to the appendix. Currently, both regression and generalized squared family processes are presented with equal detail, but shifting the finer points to the appendix and focusing more on the differences between the two settings would make the exposition more effective

3. In the experiments, the methods are evaluated solely on predictive performance, making it difficult to assess whether the multimodal capabilities of squared families are actually leveraged. Aside from Figure 4, do the authors have additional results that demonstrate the claimed benefits? The paper does not convincingly show that the unimodality of Gaussian processes is a practical limitation or that the proposed method meaningfully addresses it. A clearer argument supporting this claim would help strengthen the paper.


4. The experiments are limited in both scope, given the narrow range of datasets and baselines used, and in depth, as only NLL is reported. Additionally, it is difficult to determine whether the observed differences on these small-scale datasets reflect meaningful methodological differences or are the result of implementation issues (e.g., line 258). Do the authors have results on additional baselines or datasets, such as standard UCI regression tasks or pose estimation? Including Gaussian processes as a baseline could help clarify whether the multimodal approach is genuinely beneficial. It would also be helpful to report additional metrics, such as RMSE, to better assess performance.

I think question (1) above could be addressed if I am simply misunderstanding the paper and the authors could correct my thinking. Question (2) is certainly something that could be addressed with simple changes to the writing. Question (3/4) are also addressable but I don't know if this is something that can be done in during the short response period. Addressing 1/2 would likely me bump my score but still lean towards reject, while addressing 3/4 would lead me to lean towards weak accept. Please let me know if my criteria is not clear and I will clarify.

**Ethical Concerns:**

["NO or VERY MINOR ethics concerns only"]

**Final Justification:**

I believe the authors directly addressed my concerns for the writing changes that I suggested. However, I believe the authors did not fully address my concerns about the computational complexity since this seems a fundamental issue with the method itself. The numerical results also did not convince me of the benefits in performance or the accuracy of the comparisons. The comment I made about implementation issues (comment from line 258) was largely ignored. While I believe the paper is not yet sufficient in its comparisons I will not object if others think the paper should be accepted.

**Limitations:**

Yes.

**Quality:**

2

**Strengths And Weaknesses:**

These bullet points present the thoughts I had while reading the paper and more detailed questions / suggestions are contained in "Questions".

Strengths:
- The core idea is unique and as far as I know this particular argument of using squared families as priors is new.
- The graphics of the paper are well done and help communicate ideas. For example, Figure 4 helps give intuition about the benefit of GSPs' multimodal behavior.
- The paper's relation to previous work on squared families is present.
- The presentation contains relevant details and I believe the analytic results are accurate.

Weaknesses:
- The paper is hard to follow as things are presented without context or implicitly defined (e.g. line 14).
- The details are sometimes too much and distract from the actual point being made, see suggestion on reworking section 3.
- The numerical results are limited and make it hard to say there is any benefit to using GSFPs, see suggestion on the experiments section.

---

> ### Author Rebuttal · Authors · 2025-07-25
>
> Thanks for your detailed comments and actionable suggestions for improving the paper. See our response below. Please let us know if anything needs to be further clarified or if you have any additional questions.
>
> ---
>
> ## 1. Computational benefits of GSFs/GSPs.
>
> (same as response to Reviewer Z38J).
> We should clarify the distinction between (more general) GSFs applied to Bayesian inference and (more specific) GSFPs  applied to Bayesian inference in regression models.
>
> First, in the general case, Bayesian inference is (in)famously computationally difficult. With arbitrary likelihoods and priors, tuning computational algorithms can be practically difficult, and even fail to give good approximations unless some heavy restriction is placed on the densities (e.g. log concave). This is especially difficult when the dimension of the random variable $d$ is large, due to the curse of dimensionality, due to the exponential growth of volume. In practice, if one desires an exact computation, we need to restrict our class of probability distributions. When we restrict the class to squared families, we are able to compute exact posteriors, a thing which is usually associated with very classical models, not neural network based models. In GSFs, the complexity of this calculation is governed by the inner product of the parameter matrix $\mathbf{M}$ and the kernel matrix $K_{\mu, \mathbf{\psi}}$ (as in equation 1). Assuming the kernel matrix is known, this inner product has complexity $\mathcal{O}(n^2)$, where $n$ is the number of parameters in the GSF. Crucially, dependence of the complexity on $d$ only appears through the calculation of the kernel matrix $K_{\mu, \mathbf{\psi}}$. In practice, the kernel matrices $K_{\mu, \mathbf{\psi}}$ can be computed exactly in *linear* time in $d$. (All examples in our experiments admit linear complexity in $d$. We will enumerate more explicitly such examples in the main text --- see response to point 4 of Reviewer 7C3F). This leads to a combined complexity of $\mathcal{O}(d n^2)$, i.e. exact inference linear in dimension $d$, which is a huge improvement over inexact inference exponential in dimension $d$, which one would typically expect from numerical algorithms in general classes of probability distributions. Of course, this improvement comes with the restriction of the class of densities to GSFs, however this restriction is not too severe because GSFs are rich density approximators (as demonstrated empirically in the experiments section, as well as in previous theoretical and empirical investigations in using special cases of them as likelihood models (Rudi and Ciliberto 2021, Flaxman 2017, Walder and Bishop 2017, Loconte et al. 2023a,b, Loconte and Vergari, 2025, Loconte et al., 2024, Tsuchida et al., 2023, Tsuchida et al., 2025)). More precisely navigating the trade-off between universal approximation (making $n$ large) and computational efficiency (making $n$ small) is an important direction for future work, however we believe having $n$ as a tuneable way of navigating this trade-off is a useful perspective.
>
> Second, in the more specific case of Bayesian inference in regression models, we are only aware of two tractable (polynomial time) methods which allow for exact inference. These are GPs (Gaussian prior with Gaussian likelihood) and NGGPs (combining a GP and a normalising flow). We would also claim that these have desirable complexities when placed in the space of all possible function inference schemes. E.g. doing inference with an exponential prior on parameter weights and a Gaussian likelihood would be computationally very difficult in practice, becuase one would require numerical methods. DKT and NGGP, as well as ours, can indeed be understood as kernel methods. The complexities are similar (although NGGP has some further complexities due to the added normalising flow component). Practically, our method is slightly slower than DKT (i.e. GPs), because of the added GSF, but faster than NGGPs. See Appendix E.9. This is why we chose to highlight it as a limitation.
>
> We will add this discussion to the main paper, with the extra space afforded by extra one page as well as your suggestions from point 2 below. Please let us know if you need any clarifications.
>
> ---
>
> ## 2. Rearranging of section 3.
>
> We agree with your suggestion. In the revision, we propose the following changes. Please let us know if you think these are sufficient.
> a) Combine section 3.2 and section 3.3
> b) Most of 3.3 does not need to appear in the main paper, and will be moved to the appendix. All that is important is using a Gaussian base measure and Gausssian likelihood.
> c) Similarly, Corollary 6 can be moved to the Appendix. Corollary 7 is what matters most, as well as a claim that the marginal likelihood is also available in closed-form (with a reference to a more precise derivation in the appendix).
>
> ---
>
> ## 3. Multimodality is useful.
>
> We note that the unimodal limitation of Gaussian processes is discussed in prior work, which we cite and use as a competitor model, NGGP (Sendera et al., 2019, see second paragraph, Fig. 1, Fig. 5). Here visualisations of the failure of unimodal GPs on synthetic datasets (similar to our Fig. 4) are given. The practical limitation of GP/DKT is that it can only represent Gaussian beliefs. Any data which requires multimodal beliefs (such as our Fig. 4, or alternatively Sendera et al.'s Fig. 1, Fig. 5) will necessarily require non-Gaussian beliefs, because Gaussians are unimodal. See also our Fig. 1. We will update the manuscript to reflect this. In summary, our argument is that GPs can only represent Gaussian and therefore unimodal beliefs. Thus if a prediction may benefit from a multi-modal belief, a GP cannot realise this benefit. In contrast, a GSF is capable of representing multimodal beliefs (see Fig. 1 and Fig. 4).
>
> It is hard to visualise such multi-modal functional beliefs on real datasets (or even the datasets themselves), because we are not aware of any with 1 input dimensions. This is why we opted to add a new synthetic fig to the existing literature to demonstrate such a belief in 1 dimension. If you have any additional suggestions, please let us know.
>
> ---
>
> ## 4. Experiments.
>
> We originally considered the same few-shot learning benchmark (all datasets) as in Sendera et al., 2021. Note also that our existing QMUL dataset in section 4 is pose estimation.
>
> During the rebuttal period we applied DKT, NGGP and GSFP to vanilla regression datasets Boston, Concrete, Energy, Kin8nm, Naval, Power Plant, Protein, Wine and Yacht.
>
> Please appreciate that the rebuttal period does not allow exhaustive hyperparameter searches (kin8nm took 30 hours for NNGP, 1.3 hours for GSFP), and we will update these experiments with more time beyond the rebuttal period, or caution against general takeaways. In an attempt to simplify hyperparameter search in a way which is fair to all methods, we used the same feature extractor $\gamma$ for all methods, and shared hyperparameters across all methods. All methods share the same backbone architecture $\gamma$, a two layer MLP. The only remaining architectural hyperparameters (ignoring batch size, learning rates, etc.) are the parameter sizes $n,m$ for the squared family and the normalising flow architecture for the NGGP. We used the original NGGP paper's architecture from Sendera et al. 2021 for the normalising flow. We used $n=3$ and $m=10$ for the squared family. All datasets use an 80-20 data split. We show mean $\pm$ std of NLL
>
> | Method | Kernel | Dataset | | |  |  |  | | | |
> |:--:|:--:|:--:|:--:|:--:|:--:|:--:|:--:|:--:|:--:|:--:|
> | |  |    Boston House    |      Concrete      |       Energy       |       Kin8nm      |        Naval       |     Power Plant    |      Protein      |        Wine | Yacht  |
> |   DKL  | RBF | 0.220 ± 0.881 | 0.346 ± 0.717 | -0.298 ± 0.909 | 0.257 ± 0.693 | -1.280 ± 0.120 |   -0.041 ± 0.639   |   1.169 ± 0.765 | 1.097 ± 0.935   |   -1.784 ± 0.515   |
> | |  Spectral |    0.387 ± 1.162   |   _0.085 ± 1.565_  |  _-0.355 ± 1.039_  |   0.370 ± 0.796   |   -1.022 ± 0.198   |   -0.033 ± 0.681   |   1.178 ± 0.752   |    1.130 ± 0.723   |   -1.605 ± 0.976   |
> | | NN Linear |    0.565 ± 1.929   |    0.324 ± 1.452   |   -0.305 ± 1.036   |   0.444 ± 0.775   |  _-1.452 ± 0.391_  |   -0.043 ± 0.651   |   1.168 ± 0.711   | 1.085 ± 0.707   |   -1.317 ± 1.328   |
> | NGGP | RBF |   _0.123 ± 1.060_  |    0.743 ± 0.849   |    0.516 ± 0.516   |   0.259 ± 0.834   |    0.767 ± 0.388   |   -0.130 ± 0.312   | **0.474 ± 1.163** |   -0.825 ± 3.238   | **-3.088 ± 1.805** |
> | | Spectral |    0.361 ± 1.802   |    0.869 ± 0.468   |    0.557 ± 0.326   |   0.531 ± 0.525   |    0.562 ± 0.436   |  _-0.257 ± 0.262_  |   0.671 ± 1.236   | **-1.211 ± 4.056** |    0.525 ± 1.719   |
> | | NN Linear |    0.358 ± 1.220   |    0.543 ± 1.113   |    0.366 ± 0.450   |  _0.184 ± 0.635_  |    0.600 ± 0.428   |   -0.247 ± 0.538   |   1.278 ± 0.543   |  _-0.885 ± 4.858_  |   -0.011 ± 4.132   |
> |   GSFP   |   NN Linear  | **-0.128 ± 0.712** | **-0.150 ± 1.272** | **-1.426 ± 1.169** | **0.138 ± 0.873** | **-1.766 ± 0.180** | **-0.980 ± 1.062** |  _0.577 ± 1.030_  | 0.494 ± 1.744   | _-1.942 ± 0.941_ |
>
> No room for details here, but we also analyse signal to noise ratio of datasets. NGGP tends to perform better with higher signal to noise (protein, wine, yacht), otherwise it may overfit. GSFP still performs well on these datasets compared with DKL.
>
> MSE is exactly the Gaussian NLL (up to an additive constant) using posterior mean of $f(\mathbf{x})$ in the Gaussian likelihood. This is meaningful if the posterior can be easily summarised by its mean (e.g. a GP model). If the posterior is not easily summarised by a point predictor like in the case of GSFP or NGGP (e.g. the posterior mean may be in a low-density region between 2 modes), there is no single meaningful MSE value. Note that we report NLL under the full (possibly multimodal) posterior predictive distribution.

---

> > ### Comment · Reviewer_sKy9 · 2025-08-04
> > **Some issues addressed others not addressed**
> >
> > Thank you for your thoughtful and detailed response. I believe my point 2 is sufficiently addressed with the suggested changes in writing. I think 1,3 and 4 are partially but not fully addressed. I will increase my score to 3, but I will not object if the paper is accepted in the end.

---

### Official Review · Reviewer_2rpL · 2025-07-01

**Clarity:** 3
**Significance:** 3
**Originality:** 4
**Rating:** 5
**Confidence:** 4

**Summary:**

This paper introduces the use of squared families as priors. The authors show that when the base measure and the likelihood are conjugate, the posterior, marginal likelihood, and predictive distribution all admit closed-form expressions. The squared prior is capable of modeling multi-modal distributions, and its parameters can be parameterized by neural networks. This ability to capture multi-modality gives the method an advantage over Gaussian processes, as demonstrated in few-shot learning regression experiments.

**Questions:**

- In GSFP, the base measure is chosen to be a Gaussian with mean $\text{vec}(\mathbf{M})$, sharing the parameter $\mathbf{M}$ with the squared prior. Is this specific parameterization necessary for achieving good performance on regression tasks?
- For regression tasks, is it possible to adopt other base measures, feature maps, or likelihoods beyond the Gaussian base—for example, those mentioned in Appendix B? How robust is the method to different choices of these components?

**Ethical Concerns:**

["NO or VERY MINOR ethics concerns only"]

**Final Justification:**

This is a good paper, I think it can find application in many problems of probabilistic machine learning. My concerns regarding a more detailed literature review are also addressed in the rebuttal. So, I fully acknowledge the significance of this paper and strongly supports its acceptance.

**Limitations:**

See weaknesses and questions above.

**Paper Formatting Concerns:**

No such concerns.

**Quality:**

4

**Strengths And Weaknesses:**

Strengths
- The use of squared families as priors is quite novel. The conditions under which closed-form computations are possible are fairly general, making the method potentially applicable to a wide range of modern probabilistic machine learning problems.
- The paper is well written and easy to follow.

Weaknesses
- The paper could benefit from a more detailed literature review of recent related work on squared families, including a clearer discussion of their respective settings and how they differ from the current approach. At present, this comparison appears primarily in the early part of the introduction and is somewhat general, which may make it difficult for readers to fully appreciate the distinctions and significance of this work.

---

> ### Author Rebuttal · Authors · 2025-07-25
>
> Thanks for your review of our paper on squared family priors, and for your helpful comments for improving the paper. See our response below. Please let us know if you have any additional questions or believe something needs further clarification or elaboration.
>
> ---
>
> ## 1. Discussion and placement of related work.
>
> We will add an extended discussion of related works at the end of section 3, as follows. This builds upon the existing discussion in the early part of the introduction, but benefits from being after the technical part of the paper, so more technical details can be explained to the reader. It also further clarifies the distinction between priors/posteriors and likelihoods. Please let us know if this meets your expectation.
>
> """
>
> Here we place our contributions with respect to existing work. We reiterate that to the best of our knowledge, we are the first to consider (even special cases of) squared families as priors and posteriors. The precise nature of the results concerning likelihoods is fundamentally different to that of priors and posteriors, because for example, one does not attempt to estimate parameters in forming a posterior, one merely attempts to form a posterior belief. Nevertheless, existing work on special cases of squared family likelihoods is relevant in that expressive power, flexibility and tractability of densities is intuitively relevant to both likelihoods and posteriors, and uncertainty quantification more generally.
>
> **Kernel methods** Building off the general results of modelling non-negative functions using squared elements of RKHS [Marteau-Ferey et al., 2020], Rudi and Ciliberto [2021] model probability distributions, with tractable marginalisation and closure under multiplication. They fit likelihood models within this class by minimising a regularised L2 distance between the target likelihood and the model. While these models are nonparametric in construction, estimates resulting from minimising the objective satisfy a representer theorem type instantiation. The authors show that the model admits both favourable representational capability (in being able to represent a $\beta$-times differentiable target density with error less than $\epsilon$ with number of parameters scaling like $\mathcal{O}(\epsilon^{-r/\beta} (\log(1/\epsilon))^{r/2})$, where $r$ is the dimension of the domain of the distribution), as well as statistical estimation properties (as likelihoods, the proposed estimate converges at a rate of $N^{-\frac{\beta}{2\beta + r}} (\log N)^{r/2}$, where $N$ is the number of datapoints).
>
> **Probabilistic circuits** Probabilistic circuits [Choi et al., 2020] provide a computational framework for tractable probabilistic modelling. They are constructed as graphs of connected units, and by constraining the graph, allow for tractable marginalisation. However, they need to impose some structure on the units or the connections in order to ensure nonnegativity. Squared probabilistic circuits [Sladek, 2023, Loconte et al., 2023a,b, Loconte and Vergari, 2025] bypass this constraint by squaring the output of the circuit. Sums of squared circuits can also be instantiated [Loconte et al., 2024]. The focus on such works is typically in showing tractable (polynomial or indeed quadratic time/memory in the number of units) computation of normalising constants or marginalisation, as well as the inclusions of the function spaces imposed by different classes of probabilistic circuits.
>
> **Neural networks and finite feature models** Previous work [Tsuchida et al., 2025] has modelled likelihoods as proportional to the squared Euclidean norm of a function of the form $\Theta \psi(x)$, where $\Theta$ is a matrix and $\psi$ is a vector-valued function. Provided the features $\psi$ are rich enough, such models admit a universal approximation property, typically approximating the squared L2 distance between a square root of a target density and the model at a rate of $\mathcal{O}(1/n)$, where $n$ is a parameter count. As likelihoods, they learn target densities with a KL divergence to the target density decreasing at a rate of $\mathcal{O}(1/\sqrt{N})$, where $N$ is the number of samples. Such families of models have tractable normalising constants, Fisher information and statistical divergences. Furthermore, in some special cases, such families have not only tractable but indeed closed-form normalising constants, Fisher information and statistical divergences. An example studied previously was squared neural families [Tsuchida et al., 2023], where $\psi$ is a single hidden layer neural network. (see Appendix B and new Table in response to Reviewer 7C3F).
>
> **Applications** Poisson point processes (PPPs) are helpful variations on density modelling, where one models an intensity instead of a density. Whereas in density modelling, realisations are i.i.d. draws from a target density, in intensity modelling, realisations are conditionally i.i.d. draws from a target density given the number of realisations. The number of realisations is itself a random variable following a Poisson distribution with an expected number of points equal to the integral of the intensity over the domain.  Flaxman et al. (2017) used squared elements of RKHSs to model intensity functions, whereas Walder and Bishop (2017) used a squared Gaussian process prior to model intensity functions. Both are similar, the latter being a Bayesian extension of the former. We note that placing a prior over a function which when squared gives a intensity/density (likelihood) is completely disjoint to the problem we are solving here, which is to use a squared function as a prior density. Probabilistic circuits have been applied in converting knowledge graph embeddings into generative models [Loconte et al. 2023a]. Squared neural family likelihoods have been applied to label distribution learning [Zhang et al., 2025], which is a kind of variation on classical regression where instead of the target label being a single class, the target variable is a discrete distribution (i.e. an element of the simplex). Hence, here squared neural family models model a distribution over distributions with a closed-form normalising constant, expectation, variance, and covariance conditioned on input samples. Probabilistic modelling is leveraged to provide confidence intervals, conformal predictions, active learning, and model ensembling.
>
> """
>
> ---
>
> ## 2.  Notation overload of $\mathbf{M}$.
>
> Thanks very much for identifying this confusing typo! This was a notational overloading which we did not pick up on when preparing the draft. We do not need to tie the Gaussian base mean parameter to the parameter $\mathbf{M}$ in the squared prior. We did not tie them when considering regression tasks. We will change the symbol for the Gaussian mean parameter in the revision.
>
>
> ---
>
> ## 3. Non-Gaussian base measures and likelihoods.
>
> We will clarify in the updated manuscript.  Our claim is that the GSF priors/posteriors are rich/expressive, i.e. they can model non-Gaussian and multimodal distributions. This is not a statement about likelihoods -- indeed, we do not claim that GSFs are conjugate to all likelihoods. In the case of regression, we rely on a Gaussian likelihood and our results follow from its conjugacy with the Gaussian base measure -- but note that the priors and posteriors have a complex neural-network based non-Gaussian density with respect to that base measure (this is what makes them expressive while preserving conjugacy). We also analyse other likelihoods (e.g. Poisson in Figure 1, others in Appendix B).
>
> (Note that we did not use Gaussian feature maps $\gamma$ or $\psi$, and the feature maps do not need to be Gaussian. See table in response to 7C3F. For GSFP regression, only base measure $\mu$ and likelihood are Gaussian. More generally for GSFs, non-Gaussian base measure $\mu$ and likelihoods can be used).
>
> ---
>
> ## Additional references
>
> Zhang, D. Label Distribution Learning using the Squared Neural Family on the Probability Simplex. In The 41st Conference on Uncertainty in Artificial Intelligence. 2025

---

> > ### Comment · Reviewer_2rpL · 2025-08-04
> >
> > I thank the reviewer for the informative rebuttal. My concerns are addressed.

---

### Official Review · Reviewer_Z38J · 2025-07-01

**Clarity:** 4
**Significance:** 3
**Originality:** 4
**Rating:** 5
**Confidence:** 4

**Summary:**

This work considers the use of Generalised Squared Families (GSF) as **priors** for Bayesian linear feature regression problems.

Generalised squared densities $q(\omega | \theta, \mu, \psi)$ are defined with respect to measure $\mu$, proportionally to the squared Euclidean norm of the inner product $\theta \psi(\omega)$. Due to its analytical properties, the density can be written in terms of a ratio between two analytically tractable traces ---as in Equation (1) ---, generalizing mixture models.

The authors demonstrate in Section 3 that closed-form expressions for the marginal likelihood, posterior parameter and posterior predictive distributions are attainable for GSFs, whenever the base measure of the GSF is itself conjugate to the likelihood of the model.

Based on these analytical properties, the authors tackle Bayesian (linear in features) regression models, and showcase that for Gaussian likelihoods, Bayesian GSF priors over these models (called GSF processes) result in closed form posterior and marginals of interest.

Experiments showcase the benefits of considering a multi-modal prior/posterior family in few-shot learning examples, where better model-fit is achieved when compared to reasonable baselines.

**Questions:**

- Can the authors elaborate and more clearly explain the computational complexity of GSFPs? What are the key bottlenecks and how can one implement or overcome them?

- I would encourage the authors to discuss (if any) the potential of GSF priors in classical GP regression problems
    - what are the benefits/drawbacks of this family of priors?
    - can the authors showcase empirically how these priors would impact regression performance?

**Ethical Concerns:**

["NO or VERY MINOR ethics concerns only"]

**Final Justification:**

I recommend acceptance of this work based on my original assessment and the informative rebuttal provided by the authors.

The rebuttal addresses most of our concerns and provides new illustrative experiments to support their claims.

In addition, the authors have clarified certain aspects of the work and suggested edits to the manuscript that will improve its presentation.

**Limitations:**

Yes

**Quality:**

4

**Strengths And Weaknesses:**

**The main strength** and significant contribution of this submission is to extend Gaussian Process regression with multimodal priors and posteriors via GSFs, and to show that these have closed form solutions.

- Namely, this work can be seen as a generalization of Gaussian prior with Gaussian likelihood regression models in feature space, via GSF priors, that allow for multimodal expressions of uncertainty.

Additionally,
- The background section is a thorough and informative summary of the concepts needed to understand this work
- Section 3, where the Bayesian linear regression in feature space with GSF priors is introduced, is a math-heavy section that clearly presents all the key assumptions and properties needed to derive the closed form expressions for GSFPs.

**The main weakness**, as acknowledged by the authors themselves, is the computational (and memory) cost associated with training a GSFP.

---

> ### Author Rebuttal · Authors · 2025-07-25
>
> Thanks very much for your thoughtful review. See below for our responses to your comments and suggestions. If anything is unclear or needs further clarification, please let us know.
>
> ---
>
> ## 1. Computational complexity
>
> (Same as response to Reviewer sKy9).
> We should clarify the distinction between (more general) GSFs applied to Bayesian inference and (more specific) GSFPs  applied to Bayesian inference in regression models.
>
> First, in the general case, Bayesian inference is (in)famously computationally difficult. With arbitrary likelihoods and priors, tuning computational algorithms can be practically difficult, and even fail to give good approximations unless some heavy restriction is placed on the densities (e.g. log concave). This is especially difficult when the dimension of the random variable $d$ is large, due to the curse of dimensionality, due to the exponential growth of volume. In practice, if one desires an exact computation, we need to restrict our class of probability distributions. When we restrict the class to squared families, we are able to compute exact posteriors, a thing which is usually associated with very classical models, not neural network based models. In GSFs, the complexity of this calculation is governed by the inner product of the parameter matrix $\mathbf{M}$ and the kernel matrix $K_{\mu, \mathbf{\psi}}$ (as in equation 1). Assuming the kernel matrix is known, this inner product has complexity $\mathcal{O}(n^2)$, where $n$ is the number of parameters in the GSF. Crucially, dependence of the complexity on $d$ only appears through the calculation of the kernel matrix $K_{\mu, \mathbf{\psi}}$. In practice, the kernel matrices $K_{\mu, \mathbf{\psi}}$ can be computed exactly in *linear* time in $d$. (All examples in our experiments admit linear complexity in $d$. We will enumerate more explicitly such examples in the main text --- see response to point 4 of Reviewer 7C3F). This leads to a combined complexity of $\mathcal{O}(d n^2)$, i.e. exact inference linear in dimension $d$, which is a huge improvement over inexact inference exponential in dimension $d$, which one would typically expect from numerical algorithms in general classes of probability distributions. Of course, this improvement comes with the restriction of the class of densities to GSFs, however this restriction is not too severe because GSFs are rich density approximators (as demonstrated empirically in the experiments section, as well as in previous theoretical and empirical investigations in using special cases of them as likelihood models (Rudi and Ciliberto 2021, Flaxman 2017, Walder and Bishop 2017, Loconte et al. 2023a,b, Loconte and Vergari, 2025, Loconte et al., 2024, Tsuchida et al., 2023, Tsuchida et al., 2025)). More precisely navigating the trade-off between universal approximation (making $n$ large) and computational efficiency (making $n$ small) is an important direction for future work, however we believe having $n$ as a tuneable way of navigating this trade-off is a useful perspective.
>
> Second, in the more specific case of Bayesian inference in regression models, we are only aware of two tractable (polynomial time) methods which allow for exact inference. These are GPs (Gaussian prior with Gaussian likelihood) and NGGPs (combining a GP and a normalising flow). We would also claim that these have desirable complexities when placed in the space of all possible function inference schemes. E.g. doing inference with an exponential prior on parameter weights and a Gaussian likelihood would be computationally very difficult in practice, becuase one would require numerical methods. DKT and NGGP, as well as ours, can indeed be understood as kernel methods. The complexities are similar (although NGGP has some further complexities due to the added normalising flow component). Practically, our method is slightly slower than DKT (i.e. GPs), because of the added GSF, but faster than NGGPs. See Appendix E.9. This is why we chose to highlight it as a limitation. Our method is faster than NGGPs, the only other neural method for tractable inference in function space which we are aware of.
>
> We will add this discussion to the main paper, with the extra space afforded by extra one page as well as your suggestions  and those of other reviewers.
>
> ---
>
> ## 2. Classical regression problems
> **What are the benefits/drawbacks of this family of priors (in a classical regression setting)?**
> The second point above is pertinent. Relative to inference in general function spaces, our method is tractable, with complexity matching GPs (DKT) and NGGP. Practically, our training and inference times sit between these two methods. Empirically, due to the added flexibility of our multimodal prior/posterior, we are able to outperform GPs. Unlike NGGPs, we are also not constrained to invertible transformations of GPs. The same applies to the few-shot learning setting studied in our original submission.
>
> In terms of an intuitive justification of expressivity and therefore predictive performance in the context of **vanilla** regression, we offer the following hypothesis. We expect in general that the performance follows the ordering from worst to best
>
> GP (classical kernel) $\leq$ GP (deep kernel learning kernel) $\approx$ NGGP $\approx$ GSFP.
>
> (and recall that GP (deep kernel learning kernel) is called DKT when applied to few-shot learning rather than classical regression).
>
> The reason for this ordering is that:
>
> 1. the seminal work of DKL (Wilson et al., 2016) shows empirically that DKL outperforms classical kernels in classical regression. This leverages the expressive power of deep learning to learn more relevant priors for the task at hand than classical covariance functions. In order for a DKL model to represent a classical GP, the deep neural network representing the feature mapping just has to be the identity function. But more generally, DKL can represent non-trivial covariance functions.
>
> 2. as discussed in Appendix C of Sendera et al., the "main goal of [the original NGGP work] was to show improvement of NGGP over standard GPs in the case of a **few-shot regression task**...Intuition is that NGGP may be superior to standard GPs in a simple regression setting for datasets with non-Gaussian characteristics, but do not expect any improvement otherwise". The empirical results the authors show agreement with this intuition, showing mixed results on real data, but vastly improved results on a synthetic non-Gaussian and multimodal example. In order for the NGGP model to represent a DKT or classical GP model, the normalising flow model just has to be the identity function. But this is only valuable if the data benefits from non-Gaussian modelling. In few-shot learning, where multimodal beliefs are required (depending on the support set), this can be advantageous.
>
> 3. NGGPs push GPs through a normalising flow in order to build expressivity. The practical representational limitations of normalising flows are well-known (Stimper et al., 2022). In particular, without special tricks which do not easily apply to NGGPs, normalising flows are only capable of representing *invertible transformations* of the original base distribution (which is in this case, Gaussian) (Papamakarios et al., 2021). Therefore, if the target function of interest is not invertible, it cannot be represented by an NGGP. In contrast, squared families have no such restriction (see e.g. Tsuchida et al. (2023) for a discussion in the context of frequentist likelihoods or Rudi and Ciliberto (2021) for universal approximation but without reference to invertible transformations), and can represent distributions that normalising flows cannot. This is only valuable if the data benefits from non-Gaussian modelling.
>
> **Can the authors showcase empirically how these priors would impact regression performance (in classical regression)?**
>
> During the rebuttal period, we applied DKL, NGGP and GSFP to vanilla regression UCI datasets Boston, Concrete, Energy, Kin8nm, Naval, Power Plant, Protein, Wine and Yacht. Please see response to Reviewer sKy9 for results and discussion. These results will appear in the updated submission.
>
>
> ---
>
>
> ## Additional references
>
> Stimper, V., Schölkopf, B. &amp; Miguel Hernandez-Lobato, J.. (2022).  Resampling Base Distributions of Normalizing Flows. AISTATS
>
> Papamakarios, G., Nalisnick, E., Rezende, D. J., Mohamed, S., & Lakshminarayanan, B. (2021). Normalizing flows for probabilistic modeling and inference. JMLR

---

> > ### Comment · Reviewer_Z38J · 2025-08-04
> > **Thank you for the additional discussion and results!**
> >
> > I thank the authors for the detailed discussion on the raised concerns by me and other reviewers.
> >
> > Additionally, I appreciate the new UCI dataset experiments.
> >
> > Please include all in the updated manuscript.

---

> > > ### Author Response · Authors · 2025-08-05
> > >
> > > Thanks for your update and for helping us to improve the paper. We will be sure to include the new UCI experiments and the other promised improvements in our updated manuscript.

---

### Note · Authors · 2025-08-15

We are grateful for having a fruitful discussion with the reviewers. We summarise reviews, follow-ups and proposed changes at a high level.

Reviewers found the direction in this paper "novel", "applicable to a wide range of modern probabilistic machine learning problems", and in particular allows for a "generalization of Gaussian prior with Gaussian likelihood regression models in feature space, via GSF priors, that allow for multimodal expressions of uncertainty", showing "strong results, and comprehensively demonstrates the benefits of GSF priors".  No technical concerns surrounding accuracy of findings were raised.

All four reviewers engaged very responsibly during the discussion period. Two indicated that they increased their scores. Two reviewers give "accept" and 1 gives "borderline accept". The remaining "borderline reject" reviewer stated they "will not object if the paper is accepted in the end".

Several suggestions were made for providing more explicit information, experiments, background synthesis and clarity in the paper. We are grateful for these suggestions, and will include them in the revision. These include a more detailed discussion of computational complexity (and framing the goal of Bayesian inference more generally), discussion and placement of related work, concrete examples of closed-form kernels, explicit algorithm blocks, clarifying the separation between rich distributions and arbitrary likelihoods, as well as more isolated clarity improvements. Furthermore, it was suggested to try the method on classical regression problems (in addition to the original few-shot learning problems). We have added new benchmarks on classical regression problems, as well as a more detailed discussion about the pros and cons versus other probabilistic function inference schemes.

---

### Decision · Program_Chairs · 2025-09-17

**Decision:**

Accept (poster)

**Comment:**

Reviewers find this paper to be strong, especially the fact that its results are novel (Reviewer 2rpL, Reviewer sKy9), generalize fundamental classes of models such as mixture models (Reviewer Z38J), and that the paper is well-written with good visuals (Reviewer sKy9), and comprehensive experimental results (Reviewer 7C3F). The main weakness raised is by Reviewer 7C3F, who notes that the paper lacks a clear problem statement: I think one can view this as more of a property of the work's nature, than a weakness. Specifically, this paper is not about solving a specific problem, but rather about introducing and developing a novel general-purpose technique that can be used in many problems. In such a paper, it is important to develop the components of the technique to an adequate standard, which is exactly what the paper does in the form of deriving the key formulas for marginal likelihoods and other quantities of interest, and noting the key properties that a reader might be interested in using. There is great precedent for papers like this ending up useful and impactful over the long run, potentially in ways the authors could not have initially considered, as fundamental research in new model classes creates new kinds of tools for people to use in various ways. As consequence, I am convinced that while Reviewer 7C3F presents this as a weakness, it could simultaneously have been presented as a strength under different framing.

Given these points, and on basis of reviewer consensus, I recommend acceptance.